# TRIPSCORE: BENCHMARKING AND REWARDING REAL-WORLD TRAVEL PLANNING WITH FINE-GRAINED EVALUATION

## ABSTRACT

Travel planning is a valuable yet complex task that poses significant challenges even for advanced large language models (LLMs). However, existing benchmarks primarily equate planning ability with solving rigid constraint satisfaction problems. Solvers that excel at synthetic logic puzzles often fail to handle the ambiguity of real-world user intents. To address this, we present TripScore, a behavior-grounded benchmark and evaluation framework designed to align agent development with real-world utility. We release a large-scale dataset of 4,870 queries including 219 real-world, free-form requests for generalization to authentic user intent. We propose a unified evaluation reward that fuses feasibility and quality into a granular scalar reward. Our evaluator achieves moderate agreement with travel-expert annotations (60.75%) and outperforms multiple LLM-as-judge baselines. Leveraging TripScore, we conduct extensive experiments across diverse paradigms, including neuro-symbolic solvers, test-time search and fine-tuning. Our results reveal that while rigid solvers flounder on real-world queries, RL fine-tuning (e.g., GRPO) utilizing our unified reward significantly outperforms other methods with the same base model, effectively bridging the gap between open-source models and proprietary baselines in authentic travel planning scenarios.

## 1 INTRODUCTION

Planning is widely regarded as one of the most sophisticated cognitive skills in humans. Recently, large language models (LLMs) (Anil et al., 2023; OpenAI, 2023) have demonstrated promising capabilities in complex reasoning tasks. However, benchmarks such as meeting scheduling and graph coloring (Jimenez et al., 2024; Zheng et al., 2024; Stechly et al., 2025) reveal that planning remains a challenging domain to solve comprehensively. Among these challenges, travel planning has gained attention for its real-world impact and intrinsic complexity (Shao et al., 2024a; Chen et al., 2024).

Existing benchmarks such as TravelPlanner (Xie et al., 2024) and ChinaTravel (Shao et al., 2024a) primarily scale predefined constraints to raise task difficulty. While some recent works (Wang et al., 2025; Shao et al., 2025) have begun to assess plan quality via LLM-based preference ranking, they lack rigorous validation against human experts, and their rankings are decoupled from feasibility pass rates, making them unsuitable as direct optimization signals. Consequently, the field has over-optimized for constraint satisfaction. For instance, Hao et al. (2025) integrated LLM planning with neuro-symbolic (NeSy) solvers, dramatically boosting the constraint pass rate to 97% in TravelPlanner.

However, based on our analysis of real-world user logs, existing benchmarks have drifted far from authentic usage patterns. Our data reveals that 39.3% of users specify only minimal constraints (destinations and duration), and 38.3% submit free-form customized requests. Only 22.4% provide the detailed constraint checklists assumed by prior works. While NeSy solvers excel at the "logic puzzles" of synthetic benchmarks, they are too rigid for reality. In our experiments, these solvers suffer from exceptionally low delivery rates and scores on real-world queries, as they fail to navigate ambiguity or make necessary trade-offs. The core challenge of real-world travel planning is not solving explicit constraints, but inferring high-quality plans from vague intent.

Table 1: Comparison among TripScore and other travel-planning benchmarks. # Query denotes the number of queries in the dataset; # City denotes the number of unique cities covered by each benchmark. Abbreviations: CB = constraints-based, behav. = user behavior, req. = user request. Quality Evaluation indicates how plan quality is assessed; Output Form denotes the reported metric(s). ✓ indicates expert validation is provided; ✗ indicates that quality is not evaluated or expert validation is not provided.

| Benchmark | # Query | # City | Query Type | Query Source | Quality Evaluation | Output Form | Expert Validation |
|---|---|---|---|---|---|---|---|
| TravelPlanner (Xie et al., 2024) | 1,225 | 312 | CB Template | Synthetic | ✗ | Pass rate | ✗ |
| Trip Planning (Zheng et al., 2024) | 1,600 | 48 | CB Template | Synthetic | ✗ | Pass rate | ✗ |
| ChinaTravel (Shao et al., 2024a) | 1,154 | 10 | CB Free-Text | AI-Generated | Rule | Pass rate + Per-Dim ranking | ✗ |
| TripTailor (Wang et al., 2025) | 3,848 | 40 | CB Free-Text | AI-Generated | LLM | Pass rate + Surpass rate | ✗ |
| RealTravel (Shao et al., 2025) | 1,155 | 77 | CB Free-Text | AI-Generated | LLM | Pass rate + Preference rate | ✗ |
| **TripScore-S (ours)** | 4,593 | 821 | CB Template | Real logs (behav.) | Rule + LLM | Unified score | ✓ |
| **TripScore-R (ours)** | 277 | 134 | Free-form req. | Real logs (req.) | Rule + LLM | Unified score | ✓ |

Addressing this ambiguity requires flexible reasoning and intuition rather than rigid external solvers. Reinforcement Learning (RL) (DeepSeek-AI et al., 2025; Hao et al., 2023) offers the most promising path to endow models with this capability. But its application is bottlenecked by the lack of a high-quality, dense reward signal. Existing metrics are either too sparse (binary Pass/Fail) or too fragmented (separate preference rankings), neither of which can effectively guide point-wise policy optimization. The community urgently needs not just a real-world dataset, but a comprehensive evaluation reward that unifies feasibility and quality into a coherent signal.

In this work, we present TripScore, a behavior-grounded framework that evaluates plan feasibility along with quality and aggregates the results into a single reward score. Our main contributions are:

- **Dual-Track, Real-World Dataset.** We introduce a dual-track dataset mirrored from our real-world log analysis. The synthetic track (TripScore-S) targets the 39.3% of users providing minimal constraints (destinations and duration), while the real-world track (TripScore-R) captures the 38.3% submitting free-form user requests. Collectively, these tracks cover the majority of authentic usage patterns often ignored by prior constraint-heavy benchmarks. The dataset comprises 4,870 queries split into 3,493/158/1,219 (train/val/test). The test set includes 1,000 TripScore-S queries and 219 TripScore-R queries.

- **Comprehensive Evaluation Framework.** We propose a two-stage evaluation framework: a minimal feasibility gate, followed by a weighted reward for quality assessment. Our evaluation shows moderate agreement with travel-expert annotations (60.75%), and it outperforms a range of LLM-as-judge baselines. Crucially, this point-wise reward score enables End-to-End RL fine-tuning, allowing models to directly optimize for plan utility.

- **Extensive Experimental Analysis.** We conduct extensive experiments across diverse methods and LLMs on our benchmark, including direct prompting, test-time computation, neuro-symbolic approaches, and fine-tuning. Our results demonstrate that while NeSy approaches flounder on real-world queries, RL fine-tuning (e.g., GRPO) utilizing our TripScore reward significantly outperforms other methods with the same base model, effectively bridging the gap between open-source models and leading proprietary baselines in real-world scenarios.

## 2 RELATED WORK

**Travel planning methods.** Researchers have developed diverse approaches to address travel planning challenges. These include solver-based optimization methods Ju et al. (2024); Hao et al. (2025) and test-time compute methods Gui et al. (2025); Shao et al. (2024a); Kambhampati et al. (2024); Yang et al. (2025b). While solver-based methods achieve high success rates by adhering to rule-based constraints, they often struggle to capture nuanced user preferences. Conversely, test-time compute methods, although potentially more flexible, face limitations in real-time deployment due to their test time demands. The key to advancing travel planning while maintaining user-friendly response times lies in enhancing the reasoning capabilities of Large Language Models (LLMs). Recently, Reinforcement Learning (RL) has emerged as a promising paradigm for improving LLMs' reasoning and planning abilities (Hao et al., 2023; Shao et al., 2024b; DeepSeek-AI et al., 2025). Building on these advancements, our work investigates the integration of RL techniques into travel

planning framework, aiming to strike an optimal balance between plan quality and computational efficiency.

**Travel planning benchmarks.** Although LLMs have made significant strides in reasoning and planning capabilities, evaluation benchmarks remain far from perfect. Previous benchmarks primarily focused on domains with clear, easily quantifiable objectives, such as mathematics (Cobbe et al., 2021; Chen et al., 2023), coding and software engineering (Nguyen et al., 2025; Jain et al., 2025), web interactions (Rawles et al., 2023; Pan et al., 2024) and games (Hu et al., 2025; Paglieri et al., 2025). Recently, several benchmarks have been proposed to assess the travel plan. TravelPlanner (Xie et al., 2024) introduces commonsense and hard constraints to evaluate the feasibility of travel plans. ChinaTravel (Shao et al., 2024a) and ITINERA (Tang et al., 2024) incorporate the soft constraint to evaluate the plan quality. TripTailor (Wang et al., 2025) and RealTravel (Shao et al., 2025) further advances the field by integrating LLM-based evaluation to assess plan quality. Although these benchmarks place increasing emphasis on quality, they still lack expert judgement to verify which plans are actually superior. Our work introduces a comprehensive benchmark that unifies multifaceted evaluation criteria into a single reward, surpasses existing LLM-as-judge methods in agreement with experts. Furthermore, user queries in existing benchmarks are mostly generated under preset constraints, while we incorporate real-world large-scale user queries to assess the model's planning capabilities in complex and unpredictable scenarios. Table 1 summarizes query type/source, quality evaluation, outputs, and expert validation across benchmarks.

## 3 BENCHMARK

### 3.1 OVERVIEW

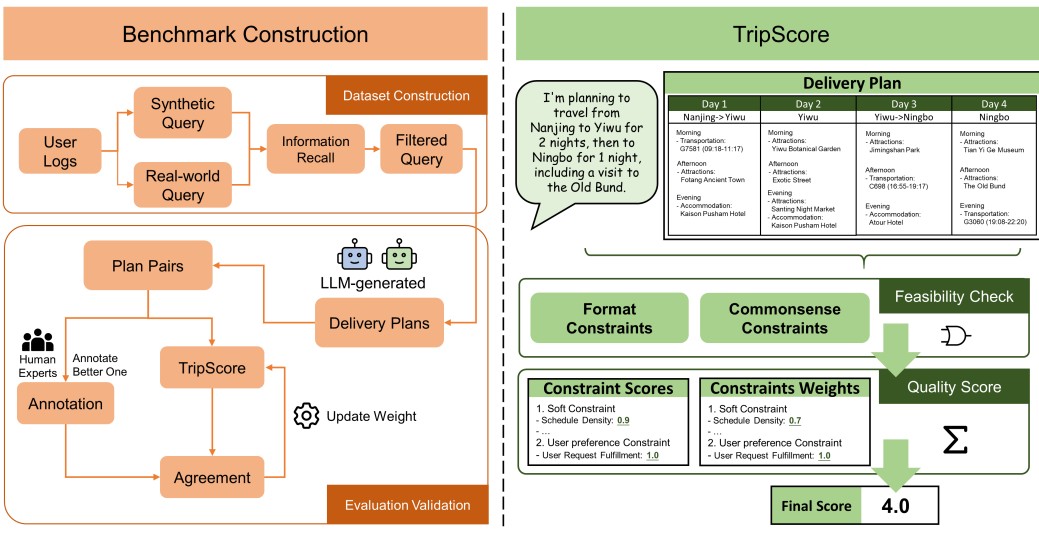

Figure 1: Construction process of TripScore dataset and illustration of TripScore.

We introduce TripScore, a comprehensive benchmark for evaluating LLMs' capabilities in complex, multi-constraint travel planning scenarios. To focus on assessing the model's core reasoning abilities, we streamline the evaluation process by directly providing all relevant information, thus eliminating extraneous tool usage complexities. This approach aligns with the sole-planning setting described in Xie et al. (2024). A representative example is illustrated in Figure 1.

Our benchmark encompasses a total of 4,870 queries, categorized into 3 splits: 3,493 training samples, 158 validation samples and 1,219 test samples. The test set contains 1,219 queries from two sources: 1,000 synthetic queries constructed from usage statistics (combinations of popular destinations and frequent durations, with randomized preferences) and 219 real-world free-form user requests. The dataset spans 897 cities, 9,376 hotels and 10,997 attractions. For a comprehensive breakdown of the dataset distribution, please refer to Appendix D.

## 3.2 DATASET CONSTRUCTION

**Real data from real users.** A key advantage of TripScore is that our data is collected from real, intrinsically motivated users, rather than paid crowd workers or AI. We developed a web application that allows users to initiate their own travel plans by selecting a destination and duration, and by providing either predefined preference constraints or free-form needs. Appendix B illustrates the main user interface and our user management policy. With explicit user consent, we log their selections and requests, recording approximately 10,000 requests per day. In our real-world logs, 39.3% of users select only destination and duration, 38.3% submit free-form customized requirements, and 22.4% choose predefined preference options. Based on this distribution, we define two tracks: one that retains only destination and duration, and another that takes free-form user requests without predefined constraints.

**Synthetic query construction.** We construct synthetic queries from user interaction logs by sampling the top 900 most requested destinations. For each destination, we attach the five most frequently chosen trip durations observed in the logs. Moreover, to capture diverse user needs, we augment queries with randomized user preferences, selected from a predefined set outlined in the personal preference constraints in Table 4.

**Real-world query construction.** To enhance authenticity, we incorporate real user requests from logs under appropriate permissions. Users submit free-form text describing their needs, and our system will return a tailored plan for them. These logs capture genuine planning intent rather than contrived prompts. These queries span a wide range, from specific tasks (e.g., attending a meeting), to day-by-day attraction preferences (e.g., Day 1: city, Day 2: zoo), and to theme-based trips (e.g., a Harry Potter trail). To maintain quality, we exclude requests under 10 words, as such inputs are overly simplistic and insufficiently challenging for models to generate tailored itineraries.

**Relevant information recall.** To ensure comprehensive and accurate retrieval, we rely on production-grade industry data curated and maintained by professional operators, with the benchmark snapshot frozen and last updated in August 2025. Time-dependent facts (e.g., operating hours) are normalized to local time and tagged with source timestamps and seasonality notes; when unavailable, constraints are marked as unknown rather than imputed. In deployment, an agent gathers relevant information via tool calls and stops once sufficient context is obtained. The planning model then consumes both the collected information and the user request to produce the final plan. However, to streamline the evaluation process in this benchmark, we bypass the tool-call stage and directly provide the planning model with the information collected by the agent. This approach allows us to focus specifically on assessing the model's planning capabilities while ensuring access to all necessary data for generating final travel plans.

**Quality control.** After collecting the user request and corresponding information, we implement a rigorous quality assurance process to remove the invalid request, leveraging both LLMs and human evaluation. We filter out requests with empty retrieved information, ensuring that each query has adequate information for planning. Then we use Gemini-2.5-flash to filter out samples whose relevant information is inconsistent with the user query. After the automated filtering step, human annotators review the remaining pool to remove requests that are prohibitively difficult or nonsensical.

## 3.3 CONSTRAINT INTRODUCTION

TripScore evaluates travel plan using four constraint types: format, commonsense, soft, and personal preferences. The first two provide a minimal feasibility check, while the latter two reflect plan quality. Format and commonsense serve as feasibility gates, while soft and personal preference scores are combined using a weighted sum to produce the final quality score. Appendix C outlines the specific constraints, and Appendix F compares constraints coverage across existing benchmarks.

**1) Format Constraint.** Format constraints ensure the structural integrity and accuracy of the generated plan. These constraints encompass a wide range of elements: strict adherence to the specified response format, rigorous information verification to prevent hallucinations, meticulous attention to detail accuracy including times and names, and ensuring the relevance of all descriptions.

**2) Commonsense Constraint.** Commonsense constraints rigorously test the model's ability to apply real-world logic and practical considerations to travel planning. These constraints are multifaceted, including: maintaining the completeness of essential information, such as not omitting necessary

round-trip transportation and hotel accommodations; ensuring a logical chronological order of activities that respects natural time progression; guaranteeing location consistency to avoid impossible movements; strictly adhering to the operating hours of attractions; and maintaining transportation consistency, such as not scheduling attractions during transit periods and ensuring consistency between arrival and departure points for subsequent transportation segments.

**3) Soft Constraint.** Soft constraints, while not strictly mandatory, play a crucial role in assessing the quality and practicality of an plan. While adherence to these constraints is not obligatory, the degree to which a plan complies correlates directly with its reward score, indicating enhanced practicality. They cover aspects such as appropriate schedule density, consistent hotel bookings, efficient use of daytime hours, avoiding repetition of attractions, and geographic clustering of locations to maximize sightseeing efficiency. Owing to their logical and arithmetic nature, we assess them with a set of rule-based metrics. Additionally, we also integrate LLMs to evaluate the plan's coverage of iconic landmarks and the diversity of attractions included, ensuring a comprehensive and well-balanced travel experience.

**4) Personal Preference Constraint.** Personal preference constraints evaluate the model's ability to tailor the plan to individual user preferences. For the synthetic dataset, the evaluation encompasses several criteria: adherence to budget expectations, alignment with desired travel pacing, prioritization of specific attraction types, and physical effort preferences. For the real-world dataset, the assessment focuses solely on whether the model fulfills specific user requests. We apply rule-based checks for the synthetic set and LLM-based judgments for the real-world set.

### 3.4 EVALUATION

Following previous work (Xie et al., 2024), we evaluate travel plans using **Delivery Rate** and **Commonsense Constraint Pass Rate**. The delivery rate calculates the ratio of plans that successfully meet all format constraints, reflecting the model's ability to understand and follow structural requirements. The commonsense constraint pass rate calculates the ratio of plans that pass all commonsense constraints among tested plans, ensuring that the generated travel plans exhibit logical consistency and real-world practicality. The Pass Rate is defined as:

$$\text{Pass Rate} = \frac{\sum_{p \in P} \mathbb{1}_{\text{passed}(p)}}{|P|}. \tag{1}$$

where $P$ represents the set of plans being evaluated by corresponding constraints, and $\text{passed}(p)$ is a function determining whether $p$ meets all the format or commonsense constraints.

Furthermore, we introduce the **Reward** for the quality of the plan, which integrates all constraints into a single metric. The final reward $\mathcal{R}$ is calculated as follow:

$$\mathcal{R}(\boldsymbol{S}; \boldsymbol{\theta}) = S_{\text{format}} + S_{\text{com}} + w_3 \frac{\sum_j w_{1,j} S_{\text{soft},j}}{\sum_j w_{1,j}} + w_4 \frac{\sum_k w_{2,k} S_{\text{pref},k}}{\sum_k w_{2,k}} \tag{2}$$

where $\boldsymbol{S} = (S_{\text{format}}, S_{\text{com}}, \boldsymbol{S}_{\text{soft}}, \boldsymbol{S}_{\text{pref}})$. $S_{\text{format}}$ and $S_{\text{com}}$ represent the format and commonsense scores. $\boldsymbol{S}_{\text{soft}}$ comprises fine-grained soft constraint scores for the individual sub-items. $\boldsymbol{S}_{\text{pref}}$ contains sub-scores for personal preference constraints. $\boldsymbol{\theta} = (\boldsymbol{w_1}, \boldsymbol{w_2}, w_3, w_4)$ are learnable weights, and the specific weight values are provided in Table 6. $j$ and $k$ denote the numbers of soft and personal preference constraint sub-items, respectively.

We adopt a lexicographic gate for hard constraints to prevent pathologically invalid plans from being rewarded: format must pass first, then commonsense. To avoid hard-penalty dominance and score compression, we bound the penalties to small constant ($S_{\text{format}} \in \{-3, +1\}, S_{\text{com}} \in \{-1, +1\}$), normalize all soft and preference sub-scores to $[0, 1]$, and combine them with weights that sum to 1. If format fails, the evaluation stops with $S_{\text{format}} = -3$; if format passes but commonsense fails, the final reward is $S_{\text{format}} + S_{\text{com}} = 0$. Otherwise, soft and preference components contribute additively. For $\boldsymbol{S}_{\text{soft}}$, each subscore follows predefined rules (Appendix I); for $\boldsymbol{S}_{\text{pref}}$, synthetic queries assess budget/pace/attraction/effort, while real-world queries use a 1-5 fulfillment score normalized to [0,1] (Appendix J). We report sensitivity to penalty constants and weight choices (Appendix E.3); results

are stable across a broad range of settings, indicating the penalties do not dominate downstream contributions.

### 3.5 EVALUATION VALIDATION

**Expert annotation.** To validate our reward score, we recruited 203 travel experts to rank in pairs of travel plans. For each query, we presented two distinct itineraries to three experts and asked them to determine which one is superior. Experts were assigned destinations with which they were familiar. In total, we obtained 1,468 route pairs with 3 annotation. The result show that inter-annotator agreement was moderate (Cohen's $\kappa = 0.5421$ for pairwise; Fleiss's $\kappa = 0.5039$ across three raters), with mean pairwise agreement of 71.69% and overall three-rater agreement of 59.00%. Further details are provided in Appendix E.1.

**Weight optimization.** We optimize the weight used in the Equation 2 based on expert annotations to compute the reward. The annotation dataset was given the final label by majority voting. Given route pairs and the ground-truth labels, we learn a scoring function $\mathcal{R}(p)$ that maximizes agreement with labels.

We formulate weight learning as:

$$\boldsymbol{\theta}^* = \arg\max_{\boldsymbol{\theta}} \frac{1}{|\mathcal{D}_{\text{train}}|} \sum_i \mathbb{I}[\text{sgn}(\mathcal{R}(p_1^{(i)}) - \mathcal{R}(p_2^{(i)})) = y^{(i)}] \tag{3}$$

where $y^{(i)}$ is the human label. $\mathcal{R}(p_1^{(i)})$ and $\mathcal{R}(p_2^{(i)})$ are the final reward scores of the plan $p_1$ and $p_2$ correspondingly. $\mathcal{D}_{\text{train}}$ is the training set.

We employ grid search optimization to find optimal weight configurations for our evaluation framework (details in Appendix E.2). To mitigate overfitting on the 1,468-pair set, we utilize three complementary diagnostics: (i) stratified 5-fold cross-validation with a nested inner-loop grid search, (ii) 1,000× bootstrap 95% confidence intervals, and (iii) correlation analyses (Kendall's $\tau$) between model score differences and human preferences. The selected weights align well with human annotations, achieving a cross-validated validation accuracy of $0.6075 \pm 0.0275$ (mean $\pm$ std) and a bootstrap accuracy of 0.6138 with a 95% CI of $[0.5967, 0.6383]$. Ordinal associations are positive on both splits (train $\tau$=0.2316, validation $\tau$=0.1892).

We contextualize the 60.75% raw agreement with a K-class symmetric-noise model. The human reliability ceiling is $r \approx 83.9\%$, and the implied latent agreement of our evaluator is $r_{\text{model}} \approx 69.5\%$, corresponding to about 82.8% of the human ceiling (Appendix E.4). This suggests the reward captures many of the factors experts use when judging plans.

**Evaluation comparison.** In this section, we compare our evaluation framework with LLM-as-judge (Zheng et al., 2023; Singh et al., 2024) under multiple backbones (Gemini-2.5-flash/pro, GPT-4o, GPT-4.1-Mini, DeepSeek-V3-0324). Table 7 reports validation agreement and Kendall's $\tau$. Across the backbones where our method is applied, it consistently attains the highest agreement (e.g., Gemini-2.5-pro: 62.62%), and exhibits stronger ordinal alignment with human preferences. Point-wise performance drops sharply once ties are included (values in parentheses), e.g. Gemini-2.5-flash 61.42 to 49.77, Gemini-2.5-pro 62.35 to 53.05, GPT-4o 53.05 to 36.15. This indicates many undecided comparisons and is consistent with weak or even negative Kendall's $\tau$. Pair-wise judging is competitive in some cases (e.g., Gemini-2.5-pro 58.77%) but requires $O(n^2)$ or $O(n \log n)$ comparisons, causing cost to grow rapidly with the number of candidates. By contrast, our framework achieves higher agreement with lower evaluation overhead.

## 4 EXPERIMENTS

### 4.1 EXPERIMENTAL SETUP

**LLMs.** Our comprehensive evaluation encompassed a diverse range of state-of-the-art models, including both proprietary and open-source LLMs. We evaluated GPT-4.1-Mini, GPT-4o (OpenAI, 2023), DeepSeek-V3-0324 (DeepSeek-AI et al., 2025), Qwen3-8B, Qwen3-14B, Qwen3-32B (Yang et al., 2025a).

**Evaluation.** We adopt the metrics of Delivery rate (DR), Commonsense constraint Pass Rate (CPR), Reward score, and the generation time to evaluate the different methods comprehensively. Because DR is a hard feasibility gate, a high correlation between DR and Reward is expected. To avoid DR dominating the interpretation, we additionally report a Conditional Reward (CondR) computed only over plans that pass all format and commonsense constraints (i.e., $DR \wedge CPR$).

Given that Table 7 shows Gemini-2.5-flash achieves the second highest agreement with human raters among candidate judges, we use it as the anchor judge in our LLM-based evaluation, considering both cost and effectiveness. And we set the temperature to 0 to ensure deterministic outputs.

**Methods.** We examined four categories of planning approaches. First, we evaluated *direct* methods, which involve the straightforward application of LLMs to the planning task. Second, we explore *test-time* compute methods, including ZS-CoT (Wei et al., 2022), ReAct (Yao et al., 2023), LLM-Modulo (Kambhampati et al., 2024), and HyperTree (Gui et al., 2025), which enhance model performance by increasing inference time. Third, we investigate the *neural-symbolic* methods, including TTG (Ju et al., 2024) and NESY (Shao et al., 2024a), which integrate LLM and symbolic method. We also assess *fine-tuning* techniques, including Supervised Fine-Tuning (SFT), Rejection Sampling Fine-Tuning (RFT) (Yuan et al., 2023), and GRPO (Shao et al., 2024b), which enhance the planning capability of models by parameter optimization. To mitigate bias from using LLM-based evaluation (Goodhart's law) and to accelerate training, we enable only the rules-based component of our evaluator for reward feedback in both RL and RFT. For implementation and training details, please refer to Appendix G.

## 4.2 Experiment Analysis

In this section, we discuss the performance of various methods and LLMs in Table 2. Our analysis reveals four critical insights regarding the trade-offs between test-time compute and model capacity mechanisms.

**The efficiency-performance trade-off in Test-Time compute.** While test-time methods (e.g., LLM-Modulo, HyperTree) generally improve plan feasibility, they incur a prohibitive latency cost. As shown in Table 2, LLM-Modulo with GPT-4o achieves the highest reward (2.69) and Delivery Rate (90.86%) in real-world settings, but this comes at the expense of tripling the inference time compared to Direct prompting (62.39s vs. 21.36s). This observation suggests distinct diminishing returns for inference-time compute in planning tasks. While iterative refinement is effective, the latency overhead scales poorly for real-time applications. The marginal gain in reward often fails to justify the linear or exponential increase in compute time.

**The rigidity of Neuro-Symbolic solvers.** Neuro-symbolic approaches (TTG, NESY) demonstrate a "Constraint Paradox", as they achieve near-perfect Constraint Pass Rates (CPR $\approx$ 99%) but suffer from exceptionally low Delivery Rates (DR) and Rewards. This result highlights the mismatch between hard-constraint solvers and real-world ambiguity. Symbolic solvers treat travel constraints as rigid logical rules, leading to "over-constrained" states where no feasible solution exists (resulting in execution failure). In contrast, end-to-end LLMs perform "soft" constraint satisfaction, finding viable trade-offs that, while not satisfying all constraints, yield executable plans (higher DR).

**For small models, test-time methods are ineffective.** A striking disparity exists between model sizes when utilizing test-time compute methods. On Qwen3-8B, advanced test-time compute methods like HyperTree not only fail to outperform but also markedly under-perform Direct prompting (Reward drops from -1.00 to -2.62). We believe that smaller models (e.g., 8B parameters) lack robust instruction-following and self-verification capabilities, making it hard to guide the planning process. Thus, providing additional test-time compute to capacity-limited models introduces noise rather than refining the final solution. Effective planning process appears to be an emergent ability that requires a stronger base reasoner such as GPT-4o.

**RL fine-tuning internalizes reasoning.** Fine-tuning methods, particularly GRPO, offer the best trade-off between plan quality and inference speed. On Qwen3-8B, GRPO achieves a Real-world Reward of 2.04, significantly outperforming Direct prompting (-1.00) and surpassing the much larger GPT-4o Direct baseline (1.61), while maintaining a latency of under 10 seconds. This validates the hypothesis that complex planning constraints can be "internalizes" as parametric knowledge. This suggests that for domain-specific planning, parameter adaptation is far more compute-

Table 2: Performance comparison of various planning approaches across different LLMs. DR and CPR are multiplied by 100. The **best** and second-best results are highlighted. Time denotes the average inference time in seconds. Values with gray shading indicate inference times exceeding 30 seconds.

| Method | LLM | Synthetic | | | | | Real-world | | | | |
|---|---|---|---|---|---|---|---|---|---|---|---|
| | | DR↑ | CPR↑ | CondR↑ | Reward↑ | Time↓ | DR↑ | CPR↑ | CondR↑ | Reward↑ | Time↓ |
| **Direct** | | | | | | | | | | | |
| Direct | GPT-4o | 77.23 | 93.73 | 2.87 | 1.40 | 20.71 | 76.71 | 80.61 | 3.73 | 1.61 | 21.36 |
| | GPT-4.1-Mini | 75.69 | 91.43 | 2.89 | 1.27 | 13.96 | 75.34 | 68.49 | 3.74 | 1.19 | 14.22 |
| | DSV3 | 73.92 | 96.00 | 2.87 | 1.26 | 27.61 | 75.78 | 78.10 | 3.66 | 1.44 | 25.03 |
| | Qwen3-8B | 27.14 | 94.98 | 3.39 | -1.31 | 9.37 | 39.73 | 56.30 | 3.61 | -1.00 | 10.48 |
| | Qwen3-14B | 39.93 | 88.70 | 3.05 | -0.72 | 12.35 | 51.59 | 67.26 | 3.49 | -0.24 | 13.15 |
| | Qwen3-32B | 42.32 | 95.13 | 2.98 | -0.53 | 19.33 | 60.27 | 71.86 | 3.88 | 0.49 | 19.83 |
| **Test-Time Compute** | | | | | | | | | | | |
| CoT | GPT-4o | 81.65 | 93.96 | 2.88 | 1.66 | 20.48 | 82.19 | 80.55 | 3.70 | 1.92 | 20.14 |
| | Qwen3-8B | 22.53 | 93.47 | 3.39 | -1.61 | 13.77 | 36.07 | 51.89 | 3.62 | -1.24 | 13.51 |
| | Qwen3-14B | 30.85 | 83.43 | 2.89 | -1.33 | 17.59 | 45.79 | 70.40 | 3.68 | -0.44 | 17.27 |
| ReAct | GPT-4o | 82.80 | 90.42 | 2.89 | 1.65 | 32.14 | 78.54 | 74.99 | 3.74 | 1.56 | 39.60 |
| | Qwen3-8B | 31.10 | 93.14 | 2.89 | -1.23 | 18.98 | 40.10 | 72.68 | 3.66 | -0.73 | 21.88 |
| | Qwen3-14B | 54.89 | 81.79 | 2.88 | -0.06 | 26.16 | 60.73 | 80.45 | 3.76 | 0.66 | 31.75 |
| LLM-Modulo | GPT-4o | 84.38 | 92.91 | 2.91 | 1.81 | 44.78 | **90.86** | 89.95 | 3.63 | **2.69** | 62.39 |
| | Qwen3-8B | 44.33 | 90.32 | 2.94 | -0.49 | 34.17 | 50.22 | 72.72 | 3.65 | -0.16 | 40.23 |
| | Qwen3-14B | 49.54 | 85.42 | 2.89 | -0.29 | 64.38 | 56.62 | 81.56 | 3.40 | 0.27 | 54.14 |
| HyperTree | GPT-4o | 81.51 | 89.39 | 2.89 | 1.55 | 49.10 | 84.01 | 70.11 | 3.73 | 1.72 | 45.96 |
| | Qwen3-8B | 14.67 | 85.75 | 2.94 | -2.19 | 31.31 | 4.56 | **100.00** | 3.80 | -2.62 | 35.92 |
| | Qwen3-14B | 27.25 | 66.92 | 2.86 | -1.66 | 34.21 | 42.01 | 63.04 | 3.85 | -0.72 | 41.71 |
| **Neural-Symbolic** | | | | | | | | | | | |
| TTG | GPT-4o | 37.10 | **99.38** | 3.60 | -0.56 | 2.21 | 12.78 | 100.00 | 3.49 | -2.17 | 2.03 |
| NESY | GPT-4o | 57.96 | 98.74 | 3.25 | 0.60 | 77.92 | 1.82 | 100.00 | 3.04 | -2.89 | 46.44 |
| | Qwen3-8B | 47.37 | 98.04 | 2.86 | -0.25 | 62.58 | 2.73 | 100.00 | 3.22 | -2.83 | 42.12 |
| **Fine-Tuning** | | | | | | | | | | | |
| SFT | Qwen3-8B | 74.02 | 91.23 | 2.88 | 1.17 | 9.00 | 77.16 | 82.84 | 3.46 | 1.53 | 11.85 |
| | Qwen3-14B | 82.80 | 93.41 | 2.86 | 1.70 | 12.69 | 85.38 | 76.47 | 3.56 | 1.89 | 15.34 |
| RFT | Qwen3-8B | 74.49 | 93.66 | 2.90 | 1.26 | 9.77 | 82.19 | 81.11 | 3.53 | 1.82 | 12.21 |
| | Qwen3-14B | 84.75 | 93.44 | 2.88 | 1.83 | 15.13 | 84.02 | 77.71 | 3.52 | 1.83 | 14.43 |
| GRPO | Qwen3-8B | 75.59 | 90.05 | 2.91 | 1.25 | 8.97 | 88.31 | 76.72 | 3.53 | 2.04 | 9.67 |
| | Qwen3-14B | **84.91** | 94.56 | 2.89 | **1.87** | 15.34 | 90.27 | 78.50 | 3.52 | 2.20 | 14.04 |

efficient than test-time search, bridging the gap between small open-source models and leading proprietary models.

### 4.3 TRAVEL PLAN QUALITY

To further disentangle feasibility from quality, we conducted a controlled pairwise comparison. We filtered out the test set to include only queries where both plans pass the feasibility check, and then compare the remaining plans head-to-head to assess the quality produced by each method. We paired the outputs of our trained models (Qwen3-14B) against the leading proprietary model (GPT-4o) and compare their plan quality. We evaluated these plans using the TripScore Reward. To avoid reward hacking, we additionally invite experts to make choices between two candidate plans and compare their quality again.

Table 3 shows that SFT, RFT and especially GRPO substantially narrow the quality gap between Qwen3-14B and GPT-4o beyond feasibility, and that these gains generalize from the synthetic dataset to the real-world dataset, validating the generalization of the fine-tuning method. For example, on the synthetic dataset under the TripScore reward, GPT-4o's preference rate decreases from 0.726 against Direct to 0.587 against SFT, 0.515 against RFT and 0.459 against GRPO. Due to the training data distribution, we observe larger improvements for GRPO on the synthetic dataset (a greater reduction in GPT-4o's win rate), but there are also consistent gains on the real-world dataset. Furthermore, There is a consistent correlation between TripScore Reward and Expert Selection. Expert evaluation corroborates this trend, confirming that GRPO-tuned small models can match the quality of leading proprietary models.

Table 3: Pairwise comparison on plan quality only (identical queries per pair). $H_0$: $p = 0.5$ (two-sided). 95% Wilson CIs shown for A (left model).

| Test Set | Pair (A vs B) | A Win Rate | 95% CI | $p$-value |
|---|---|---|---|---|
| **TripScore Reward** | | | | |
| Synthetic | Direct (GPT-4o) vs Direct (Qwen3-14B) | 0.726 | [0.614, 0.845] | 0.002 |
| | Direct (GPT-4o) vs SFT (Qwen3-14B) | 0.587 | [0.421, 0.686] | 0.632 |
| | Direct (GPT-4o) vs RFT (Qwen3-14B) | 0.515 | [0.352, 0.675] | 1.000 |
| | Direct (GPT-4o) vs GRPO (Qwen3-14B) | 0.459 | [0.284, 0.639] | 0.484 |
| Real-world | Direct (GPT-4o) vs Direct (Qwen3-14B) | 0.744 | [0.589, 0.854] | 0.004 |
| | Direct (GPT-4o) vs SFT (Qwen3-14B) | 0.659 | [0.505, 0.784] | 0.032 |
| | Direct (GPT-4o) vs RFT (Qwen3-14B) | 0.552 | [0.375, 0.716] | 0.711 |
| | Direct (GPT-4o) vs GRPO (Qwen3-14B) | 0.522 | [0.486, 0.557] | 0.245 |
| **Expert Selection** | | | | |
| Synthetic | Direct (GPT-4o) vs Direct (Qwen3-14B) | 0.701 | [0.604, 0.814] | 0.004 |
| | Direct (GPT-4o) vs SFT (Qwen3-14B) | 0.549 | [0.479, 0.586] | 0.137 |
| | Direct (GPT-4o) vs RFT (Qwen3-14B) | 0.504 | [0.418, 0.581] | 0.839 |
| | Direct (GPT-4o) vs GRPO (Qwen3-14B) | 0.477 | [0.365, 0.603] | 0.628 |
| Real-world | Direct (GPT-4o) vs Direct (Qwen3-14B) | 0.785 | [0.671, 0.829] | 0.002 |
| | Direct (GPT-4o) vs SFT (Qwen3-14B) | 0.614 | [0.572, 0.654] | 0.008 |
| | Direct (GPT-4o) vs RFT (Qwen3-14B) | 0.561 | [0.409, 0.682] | 0.636 |
| | Direct (GPT-4o) vs GRPO (Qwen3-14B) | 0.513 | [0.478, 0.548] | 0.492 |

## 4.4 DISTRIBUTION ANALYSIS OF ERROR TYPES

To analyze the pass rate, Figure 2 breaks down violations of format and commonsense constraints on the real-world set. For clarity, we report the top 5 most frequently broken constraints for each method. And we found consistent dominant errors across all methods are format: information accuracy and response format, and commonsense: operating hours and chronological order. Notably, Qwen3-8B exhibits the highest proportion of information-accuracy failures. These typically manifest as hallucinations, such as mismatched attraction IDs and names. Both SFT and GRPO training methodologies demonstrate significant efficacy in mitigating this type of hallucinations.

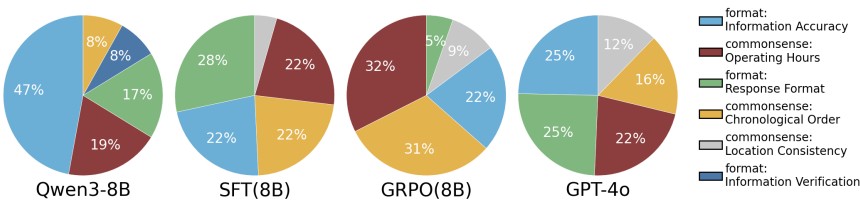

Figure 2: Format and commonsense constraints error distribution in the real-world test set.

## 4.5 PERFORMANCE ON VARIOUS TRIP DURATIONS

Figure 3 model performance as task complexity increases (longer trip durations). To maintain statistical significance, we focus on results for trips ranging from 1 to 5 days, excluding longer durations with insufficient data points. As the trip duration extends from 1 to 5 days, we observe a notable degradation in performance across vanilla baselines. This decline is evident in both format constraint (Average DR) and commonsense constraint (Average CPR), with a corresponding decrease in reward. In contrast, GRPO (14B), which denotes the GRPO method applied to the Qwen3-14B model, generally achieves the highest DR and CPR scores while exhibiting the smallest performance decline as trip duration increases, indicating superior stability in long-horizon planning scenarios. This suggests that GRPO improves the model's reasoning ability for complex itineraries: it learns to maintain the consistency and constraint satisfaction (e.g., operating hours and chronological order) while preserving valid output structure, leading to stronger performance when scheduling becomes more challenging.

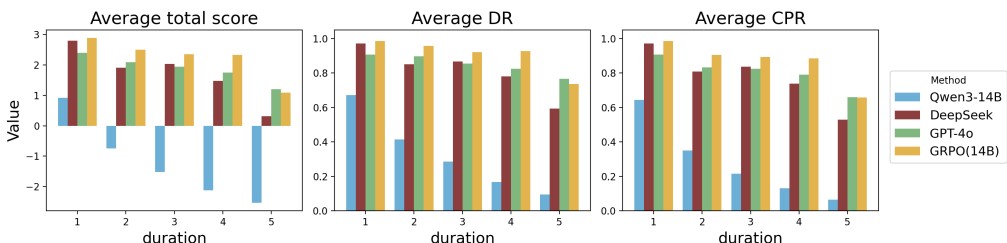

Figure 3: Performance distribution across varying trip durations in the synthetic test set.

## 4.6 CASE STUDY

To examine how plan quality is assessed, we present several head-to-head cases in Figure 4. Although all plans are feasible, side-by-side comparison under our criteria clearly exposes which route is superior. This underscores the need for fine-grained comparative evaluation rather than relying solely on rules or single-pass LLM judgments. Reliable evaluation must account for structural validity, spatio-temporal coherence, semantic value (iconicity and diversity), and user preferences. A practical evaluator should fuse rule-based diagnostics with preference signals, quantify uncertainty, and return actionable feedback.

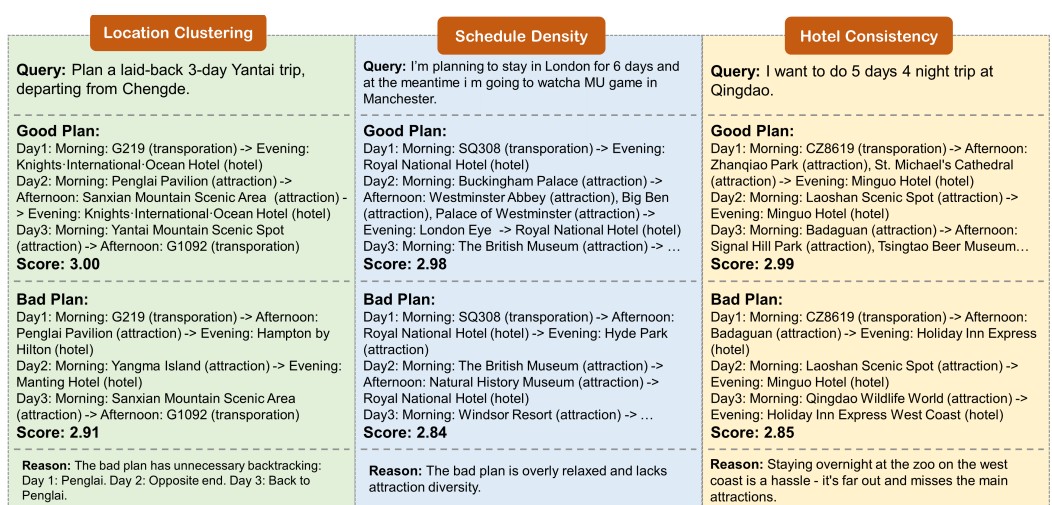

Figure 4: Case studies of the comparison of the plan quality.

## 5 CONCLUSION

In this work, we introduce a comprehensive evaluation framework that aggregates fine-grained criteria into a single reward for plan quality, achieving 60.75% agreement with expert annotations and surpassing LLM-as-judge baselines. And we also present a large-scale, real-world dataset that encapsulates authentic user requests, providing a rich resource for developing and testing travel planning systems. Finally, we conduct extensive comparative experiments that underscore the effectiveness of our framework and demonstrate the promise of fine-tuning techniques for improving plan quality while maintaining low inference time.

## ETHICS STATEMENT

**User Data Authorization.** All user data utilized in this study was obtained with explicit authorization from the users. We affirm that this data is used solely for academic research purposes and will not be employed for any commercial applications.

## REPRODUCIBILITY STATEMENT

All evaluation and experimental code is provided in the supplementary material.

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

## A    THE USE OF LARGE LANGUAGE MODELS (LLMs)

In the preparation of this manuscript, large language models were employed exclusively for the purpose of writing refinement and stylistic enhancement. These models were not used for generating research ideas, conducting analyses, or drawing conclusions.

## B    WEB APPLICATION USAGE

Our web application interface is designed for simplicity, ensuring effortless access for users worldwide. Figure 5 showcases the main screen through which users submit their travel requests and generate travel plans. Users can select destinations and duration, choose either enumerated preferences or free text needs. Then users can click on "Plan a Trip with AI" to generate tailored travel plans.

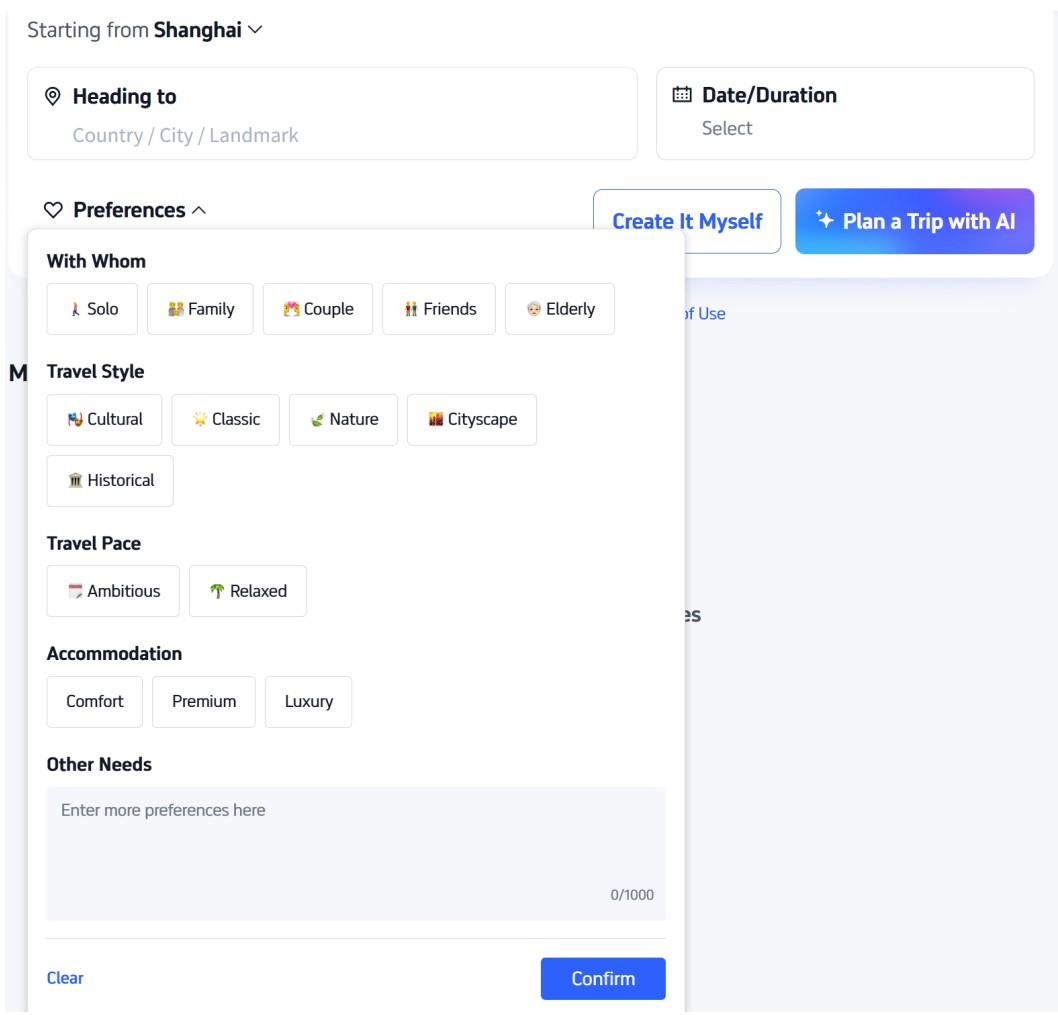

Figure 5: Main interface of the travel-planning web application used for collecting real-user interaction logs.

To mitigate the risk of collecting low-quality data, we enforce several quality-control measures. First, users must register with an email or phone number for identity verification. Second, we audit logs in real time and take action to ban accounts that generate NSFW content, open multiple sessions or operate at abnormal speed. Third, we use an NSFW keyword list blocks to prevent users from harmful outputs. Last, each user is initially allowed at most 100 interactions per day.

## C  CONSTRAINTS

As illustrated in Table 4, we evaluate the feasibility and quality of a plan from four dimensions: format constraint, and commonsense constraint for feasibility check, and soft constraint and personal preference constraint for quality evaluation.

Table 4: Constraint description. The format constraints assess whether the plan adheres to the specified format guidelines. The commonsense constraints evaluate the logical feasibility of the plan. The soft and preference constraints measure the alignment of the plan with plan quality standards and user preferences. Constraints in black originate from existing benchmarks, whereas constraints in blue are our newly proposed constraints.

| Constraint | Description | Evaluation |
|---|---|---|
| *Format Constraint* | | |
| Response Format | All responses must follow the requested structure and organization exactly as specified. | Rule |
| Information Verification | All attractions, transportation, and hotels in the plan must come from the provided information, otherwise, it will be considered a hallucination. | Rule |
| Information Accuracy | Details like name and departure time must match the provided information. | Rule |
| Information Relevance | All descriptions must specifically match their intended attractions (e.g., not mixing up or blending details from different places). | Rule |
| *Commonsense Constraint* | | |
| Information Completeness | All necessary information must be included, especially accommodation for each destination with multi-day stay and essential transportation. | Rule |
| Chronological Order | All activities must be listed in chronological order. | Rule |
| Location Consistency | Each day's activity must be scheduled in the city where the traveler is actually present and change only after any required transportation. | Rule |
| Operating Hours | Visits to attractions must only be scheduled during their confirmed opening hours. | Rule |
| Travel Block-Out | No activities can be scheduled between departure and arrival times during transportation. | Rule |
| Transport. Consistency | Arrange the transportation from the starting city to each destination in sequence to avoid jumps or repetitive routes. | Rule |
| *Soft Constraint* | | |
| Schedule Density | Each day's plan should be thoughtfully paced, ensuring neither overly lengthy nor excessively brief periods of activity. | Rule |
| Hotel Consistency | When staying in the same city, the same hotel should be used throughout to avoid unnecessary check-ins and check-outs. | Rule |
| Daytime Utilization | Fill daytime hours when evening travel is planned; avoid starting the day with attractions that open late. | Rule |
| Unique Attractions | Attractions should not appear more than once in the plan. | Rule |
| Location Clustering | When possible, group attractions that are close to each other to reduce travel time and maximize sightseeing time. | Rule |
| Iconic Landmarks | Include renowned local attractions and must-see sites in the plan. | LLM |
| Attraction Diversity | Avoid overrepresentation of similar attractions to ensure a varied and engaging itinerary. | LLM |
| *Personal Preference Constraint* | | |
| Budget Preference | The plan should align with the user's budget expectations (eg. "premium", "budget-conscious", or "best value"). | Rule |
| Pacing Preference | The plan should reflect the user's desired pacing (eg. "relaxed", "moderate", or "compact"). | Rule |
| Attraction Prioritization | The plan should prioritize or include the user-specified types of attractions. | Rule |
| Physical Effort Preference | The plan should balance walking distances and physical intensity, matching the user's indicated effort level (eg. "light", "moderate", or "strenuous"). | Rule |
| User Request Fulfillment | The plan should follow the user's specific request. | LLM |

## D  BENCHMARK DETAILS

In Table 5, we list the detailed data distribution on training, validation and test set.

## E  EXPERT ANNOTATION

### E.1  EXPERT ANNOTATION

We conducted a comprehensive comparative evaluation, integrating expert assessments with our automated framework. We employed DeepSeek-V3, Qwen3-32B and GPT-4 to independently generate travel plans based on 3,000 user queries. After filtering out plans that failed to meet format and commonsense constraints, we obtained 1,468 final pairwise comparisons: 267 between Qwen3-32B

Table 5: Statistics of dataset distribution. We divide the dataset into training, validation and test splits and calculate the entity number correspondingly.

| Dataset | Type | #Examples | #City | #Hotel | #Attraction |
|---------|------|-----------|-------|--------|-------------|
| Training | Synthetic | 3,493 | 781 | 7,830 | 9,788 |
| Validation | Synthetic | 100 | 70 | 1,057 | 1,525 |
|  | Real-world | 58 | 41 | 638 | 1,313 |
| Test | Synthetic | 1,000 | 358 | 4,459 | 5,452 |
|  | Real-world | 219 | 122 | 1,894 | 2,057 |
| **All** |  | 4,870 | 897 | 9,376 | 10,997 |

and DeepSeek-V3, 303 between Qwen3-32B and GPT-4o, and 898 between GPT-4o and DeepSeek-V3. Then a panel of 203 travel specialists was tasked with ranking these pairs of travel plans and providing rationales for their choices. For each pair, the specialists had three options: favor route A, favor route B, or neither route met satisfactory standards. To ensure the reliability of the labeling results, each specialist was assigned routes featuring destinations with which they were familiar. And each plan pair was independently evaluated by three expert annotators.

We computed inter-annotator agreement using Cohen's $\kappa$ for pairwise comparisons and Fleiss' $\kappa$ for multi-rater reliability. The results demonstrated moderate agreement among the experts, with Cohen's $\kappa$ reaching 0.5421 for pairwise comparisons and Fleiss' $\kappa$ achieving 0.5039 for multi-rater reliability. Furthermore, we examined the agreement from different perspectives: the average pairwise agreement between annotators was 71.69%, while the overall agreement across all three raters was 59.00%.

### E.2 GRID SEARCH

We employ a grid search optimization to find the optimal weight for our evaluation framework. The search space consists of 13 parameters: 7 soft constraint weights ($w_1$), 4 preference weights ($w_2$), and 2 multiplier weights ($w_3, w_4$). We discretize the parameter space using coarse grids: $w_1 \in \{0.1, 0.4, 0.7\}$, $w_2 \in \{0.2, 0.6, 1.0\}$, $w_3 \in \{0.8, 1.0, 1.2\}$, and $w_4 \in \{0.1, 0.4, 0.6, 0.8, 1.0, 1.2, 1.4\}$. The optimized weight values are provide in Table 6.

We conduct stratified 5-fold cross-validation with a nested inner-loop grid search. The outer-fold validation accuracies are $[0.5933, 0.6049, 0.6185, 0.5743, 0.6466]$, yielding a mean of $0.6075 \pm 0.0272$ (mean $\pm$ std). On the full 1,468-pair set, $1{,}000\times$ bootstrap gives an accuracy of 0.6138 with a 95% CI of $[0.5967, 0.6383]$. Correlation analysis shows a positive but weak ordinal association between our model's score differences and human preferences: train set Kendall's $\tau=0.2316$; validation set Kendall's $\tau=0.1892$.

### E.3 SENSITIVE ANALYSIS

Moreover, we probe robustness by jointly scaling the penalty multipliers $(w_3, w_4) \in \{(0.5, 0.05), (1.0, 0.10), (1.5, 0.15), (2.0, 0.20)\}$ and by applying simplex normalization (including temperature smoothing) to the soft and preference constraint weights. Across all variants, train agreement remains in the range 59.40%-62.26% and validation agreement in 58.47%-61.32%, i.e., within $\approx 2.8$ pp (train) and $\approx 2.9$ pp (val) absolute of the baseline. The near-constant validation accuracy under multiplier scaling and simplex reparameterizations indicates the method is robust to these design choices and does not rely on a narrow hyperparameter configuration.

### E.4 NOISE-ADJUSTED ANALYSIS

Under a standard K-class symmetric-noise model, we can relate the average human-human pairwise agreement $A_{\text{pair}}$ and a typical annotator's agreement with respect to the latent truth $r$ as follows:

$$A_{\text{pair}} = r^2 + \frac{(1-r)^2}{K-1} \tag{4}$$

Table 6: Optimized weights value. Due to the different evaluation method, synthetic and real-world datasets share the weights for soft constraints but utilize distinct preference weights.

| Parameter | | Synthetic | Real-world |
|---|---|---|---|
| **w1** (soft constraints) | Schedule Density | | 0.70 |
| | Hotel Consistency | | 0.50 |
| | Daytime Utilization | | 0.40 |
| | Unique Attraction | | 0.20 |
| | Location Clustering | | 0.70 |
| | Iconic Landmark | | 0.10 |
| | Attraction Diversity | | 0.20 |
| **w2** (preferences) | Attraction Prioritization | 0.20 | - |
| | Pacing | 0.60 | - |
| | Budget | 0.60 | - |
| | Physical Effort | 0.60 | - |
| | User Request | - | 1.00 |
| **w3** (soft constraint multiplier) | | 1.00 | |
| **w4** (preference multiplier) | | 0.10 | 1.40 |

$$r = \frac{1 + \sqrt{(K-1)(K \cdot A_{\text{pair}} - 1)}}{K} \tag{5}$$

Where $K$ is the number of classes. With $K = 3$ and $A_{\text{pair}} = 0.7169$, we calculate the human reliability ceiling, $r \approx 0.8390$. Then the expected model-human agreement $A_{\text{model},h}$ relates to the model's latent-truth agreement $r_{\text{model}}$ by:

$$A_{\text{model},h} = r_{\text{model}} \cdot r + \frac{(1 - r_{\text{model}})(1 - r)}{K - 1} \tag{6}$$

Which can be rearranged to solve for $r_{\text{model}}$:

$$r_{\text{model}} = \frac{K \cdot A_{\text{model},h} - A_{\text{model},h} + r - 1}{K \cdot r - 1} \tag{7}$$

Substituting $A_{\text{model},h} = 0.6075$, $K = 3$ and $r \approx 0.8390$ yields $r_{\text{model}} \approx 0.695$. Thus, the raw 60.75% agreement corresponds to a noise-adjusted latent agreement of approximately 69.5%. We can express this as a ratio of the model's performance to the human reliability ceiling $r_{\text{model}}/r \approx 0.828$. This indicates that the model achieves about **82.8%** of the human reliability ceiling.

As a verification, we can also calculate the predicted three-annotator all-agree rate $A_{\text{all}}$ by the calculated human reliability ceiling $r$:

$$A_{\text{all}} = r^3 + \frac{(1 - r)^3}{(K - 1)^2} \approx 0.594 \tag{8}$$

This closely matches the observed overall agreement of 0.590 between all annotators, providing a sanity check for our calculations.

### E.5 EVALUATION COMPARISON EXPERIMENTS

We implement two LLM-as-judge paradigms to evaluate pairs of candidate travel plans and evaluate agreement with expert labels as well as rank correlation (Kendall's $\tau$). Both paradigms use a structured rubric of hard/soft constraints and explicitly condition on the user request. To improve reproducibility, we use frozen prompts, temperature 0, and report results across 3 runs. And we report the results for multiple LLMs (GPT-4o, GPT-4.1-Mini, DeepSeek-V3-0324, Gemini-2.5-flash/pro).

**Point-wise scoring.** The LLM independently scores each plan in $[0, 100]$ under the same request and rubric. To mitigate instability reported in prior work (Bai et al., 2023), we adopt comparative prompting, anchoring scores with comparative references. The predicted winner is the plan with the

higher score; ties yield a neither decision. We report two point-wise metrics: tie-excluded accuracy and tie-inclusive accuracy. The full prompt is provided in Appendix K.6.

**Pair-wise comparison.** The LLM receives the request and both travel plans, first eliminates candidates with hard-constraint violations, then compares soft quality and preference matching; if still uncertain, it selects the clearer, more executable plan. The model must output exactly one token from route A, route B. The prompt is provided in Appendix K.7.

**Ours scoring method.** For our rule-and-LLM-hybrid evaluation framework, we have the same setting as point-wise scoring: the predicted winner is the plan with the higher score, and ties yield neither decision. We employ various LLMs for the LLM-based component. Because tie-excluded and tie-inclusive accuracies are nearly identical, we report only the tie-inclusive accuracy in Table 7.

Table 7: Comparison with human annotations for our method and LLM-as-judge baselines. For point-wise results, values in parentheses indicate tie-inclusive accuracies.

| Method | LLM | Accuracy | Kendall's $\tau$ |
|---|---|---|---|
| Point-wise | Gemini-2.5-flash | 61.42 (49.77) | 0.2182 |
| | Gemini-2.5-pro | 62.35 (53.05) | 0.2347 |
| | GPT-4.1-Mini | 47.01 (34.74) | -0.0224 |
| | GPT-4o | 53.05 (36.15) | 0.0466 |
| | DeepSeek-V3 | 51.02 (29.76) | 0.0193 |
| Pair-wise | Gemini-2.5-flash | 57.94 | 0.1529 |
| | Gemini-2.5-pro | 58.77 | 0.1675 |
| | GPT-4.1-Mini | 49.53 | -0.0795 |
| | GPT-4o | 53.27 | 0.0762 |
| | DeepSeek-V3 | 51.64 | 0.0104 |
| Ours | Gemini-2.5-flash | 60.75 | 0.1892 |
| | Gemini-2.5-pro | **61.32** | 0.2124 |
| | GPT-4o | 57.94 | 0.1488 |
| | DeepSeek-V3 | 56.48 | 0.1196 |

# F  CONSTRAINTS COMPARISON

Constraints across benchmarks are summarized in Table 8, showing which dimensions are evaluated and by what mechanism.

Table 8: Dimension-wise coverage of evaluation constraints across benchmarks. Cells indicate whether a constraint is evaluated and by which mechanism (Eval: Rule or LLM). ✓ = evaluated; ✣ = similar constraint exists; ✗ = not evaluated.

| Dimension | Constraints | Ours (Eval) | ChinaTravel (Eval) | TravelPlanner (Eval) | TripTailor (Eval) |
|---|---|---|---|---|---|
| Format | Response format | ✓(Rule) | ✓(Rule) | ✓(Rule) | ✓(Rule) |
| | Info verification | ✓(Rule) | ✓(Rule) | ✓(Rule) | ✓(Rule) |
| | Info accuracy | ✓(Rule) | ✓(Rule) | ✓(Rule) | ✓(Rule) |
| | Info relevance | ✓(Rule) | ✗ | ✗ | ✗ |
| Commonsense | Info completeness | ✓(Rule) | ✓(Rule) | ✓(Rule) | ✓(Rule) |
| | Chronological order | ✓(Rule) | ✓(Rule) | ✗ | ✗ |
| | Location consistency | ✓(Rule) | ✓(Rule) | ✓(Rule) | ✓(Rule) |
| | Operating hours | ✓(Rule) | ✓(Rule) | ✗ | ✗ |
| | Travel block-out | ✓(Rule) | ✗ | ✗ | ✗ |
| | Transport consistency | ✓(Rule) | ✓(Rule) | ✓(Rule) | ✓(Rule) |
| Soft | Schedule density | ✓(Rule) | ✗ | ✗ | ✗ |
| | Hotel consistency | ✓(Rule) | ✗ | ✗ | ✗ |
| | Daytime utilization | ✓(Rule) | ✗ | ✗ | ✗ |
| | Unique attractions | ✓(Rule) | ✣(Rule) | ✗ | ✣(Rule) |
| | Location clustering | ✓(Rule) | ✣(Rule) | ✗ | ✣(Rule) |
| | Iconic landmarks coverage | ✓(LLM) | ✗ | ✗ | ✗ |
| | Attraction diversity | ✓(LLM) | ✗ | ✗ | ✗ |
| Preference | Budget style | ✓(Rule) | ✣(Rule) | ✣(Rule) | ✣(Rule) |
| | Pacing style | ✓(Rule) | ✗ | ✗ | ✗ |
| | Attraction prioritization | ✓(Rule) | ✗ | ✗ | ✗ |
| | Physical effort | ✓(Rule) | ✗ | ✗ | ✗ |
| | User request | ✓(LLM) | ✗ | ✗ | ✓(LLM) |

# G EXPERIMENT DETAILS

## G.1 FINE TUNING

**SFT.** We fine-tune the pretrained model with pairwise query-response supervision. The SFT training set is constructed by prompting GPT-4o to answer queries from the synthetic training split. Then we filter out samples that violate our format or commonsense constraints. This yields 2,094 training and 71 validation examples. As GPT-4o does not expose an explicit chain-of-thought, we train the Qwen3 base model in a non-thinking configuration. During training, we set the maximum sequence length to 115,000 tokens and training epochs to 3. As the context length is too long, which may lead to GPU out of memory, we set the sequence parallel to 8 to ensure training progress.

**RFT (Yuan et al., 2023).** We further refine the SFT-tuned Qwen3 models via RFT. Starting from the queries in the synthetic training split, we prompt the SFT model to generate five candidate routes per query and score each route with our own evaluator. The highest-scoring route is retained; queries for which all five candidates fail the quality bar are dropped. This curation leaves 2,372 clean training samples. We then fine-tune the SFT model for three epochs on this set, using the same hyperparameters as SFT.

**GRPO (Shao et al., 2024b).** The detailed explanation of GRPO is provided in Appendix H. We train the SFT model based on GRPO algorithm on queries from the synthetic training set. Given that the SFT model was trained without a thinking mode, we preserve this setting and continue fine-tuning it using the GRPO algorithm. For each query, the policy model generates 8 rollouts, and rewards are computed by our evaluation framework. To speed up training, we use only the rule-based components of our evaluator. During training process, the maximum prompt length is 79000 and the maximum answer length is 7000.

We perform early stopping by selecting the best-performing checkpoint for each task independently. The hyperparameters employed in fine-tuning baselines are presented in Table 9.

Table 9: The hyperparameters we employ in baselines.

| Method | Hyperparameter | value |
|---|---|---|
| SFT | learning rate | 1e-5 |
| | scheduler type | cosine |
| | batch size | 1 |
| | training epoch | 3 |
| | warmup ratio | 0.1 |
| RFT | learning rate | 1e-5 |
| | scheduler type | cosine |
| | batch size | 1 |
| | training epoch | 3 |
| | warmup ratio | 0.1 |
| GRPO | actor learning rate | 1e-6 |
| | scheduler type | constant |
| | batch size | 24 |
| | training epoch | 1 |
| | rollout temperature | 1.2 |
| | rollout times | 8 |

For deployment, we leverage vLLM (Kwon et al., 2023) to serve both Qwen3-8B, Qwen3-14B and Qwen3-32B. To maximize inference throughput, we configure tensor parallelism to 8 and run the models on a single 8*A100 node.

## G.2 OTHER BASELINES

**Direct.** This approach involves inputting the query directly into the model, accompanied by comprehensive instructions detailing the task requirements and all relevant gathered information. The model is expected to generate a response based solely on this input. For Qwen3 base models, we disable the thinking mode for the direct method.

**Zero-Shot Chain-of-Thought (ZS-CoT) (Wei et al., 2022).** This method enhances the reasoning process by encouraging the model to articulate intermediate steps. Building upon the Direct method, we augment the prompt with the phrase "Let's think step by step." This addition is designed to elicit a more detailed, structured reasoning process from the model, potentially leading to more accurate and transparent outcomes. For Qwen3 base models, we enable the thinking mode for the ZS-COT method.

**ReAct (Yao et al., 2023).** This method incorporates environmental feedback into the planning process. We provide the function FeasibleEnquiry to check day-level feasibility to help enhance the model's reasonning ability. We develop specialized prompts to guide the LLM in generating daily travel plans and refining them based on the function feedback. Once the model calls the Finish function, the latest plan is returned as the final output. To controll the time cost, LLMs are allowed to call the function up to 5 times; otherwise, it is regarded as failing to generate the plan.

**LLM-Modulo (Kambhampati et al., 2024).** LLM-Modulo is a hybrid method where a large language model (LLM) generates structured representations from a user's request, and a symbolic planner verifies them and flags issues in a feedback loop. The method benefits from the bidirectional interaction: The symbolic module actively gives feedback to the LLM, and the LLM refines the itinerary iteratively.

To implement the LLM-Modulo into our benchmark, we develop specialized prompts to guide the LLM in generating initial itineraries and refining them based on evaluator feedback. We reuse our existing constraint evaluators to automatically score itineraries and provide structured feedback to the LLM, and for fairness we only utilize the rule-based part in our evaluation framework. We also add a controller that coordinates iterations between the LLM and the evaluators until constraints are satisfied or a timeout is reached. We set a maximum of 3 iteration steps and stop the refinement process if the reward score exceeds 3.5.

**HyperTree (Gui et al., 2025).** This baseline employs a hypertree-structured reasoning paradigm that recursively decomposes a travel query into transportation, accommodation, and attraction subtasks. By employing this recursive decomposition strategy, the baseline ensures that each aspect of the travel plan can be meticulously crafted and seamlessly integrated, resulting in a highly personalized and adaptable itinerary.

To adapt HyperTree to our dataset, we implement several modifications. We remove all the hypertree rules and nodes related to Dining to align with our schema. We also refactor the hypertree library to handle an arbitrary-length destination list, dynamically generating city-specific subtrees and their corresponding inter-city transportation segments, with no upper bound on the number of cities. Additionally, as the repository does not provide prompts, we write them from scratch based on the descriptions in the paper.

**TTG (Ju et al., 2024).** TTG (To The Globe) is a hybrid travel-planning system that translates natural language requests into structured JSON constraints using a fine-tuned LLM. It then solves these constraints with a Mixed Integer Linear Programming (MILP) solver to guarantee feasible and near-optimal itineraries in under 5 seconds. It combines the LLMs' natural language abilities with the mathematical guarantees of MILP solvers.

To implement TTG into our benchmark, we define a symbolic type to represent feasible travel constraints, serving as the intermediate format for MILP problem generation. Then we translate travel requests into MILP formulations that encode itinerary feasibility and optimization objectives. Finally, we integrate the Python `PuLP` library to solve the constructed MILP problems using appropriate solver strategies.

**NESY (Shao et al., 2024a).** The NESY baseline scheme consists of a two-stage process: (1) In the NL2DSL translation stage, natural language queries are converted into logical and preference DSL requirements; (2) In the interactive search stage, a neuro-symbolic solver sequentially arranges activities under the guidance of a symbolic sketch and LLM-driven POI recommendations, generating a multi-day itinerary with DSL validation.

To adapt NESY to our dataset, we implement the following modifications:

- **Dataset Adaptation**. We streamline the planning process by removing redundant logical modules irrelevant to our study, such as restaurants, ensuring the workflow aligns with the available data support.

- **DSL and Concept Function Reconstruction**. We reconstruct the concept functions and augmented the DSL statements with commonsense constraints to match the specific format of our itinerary results. Specifically, we removed DSL statements related to restaurants, people number, dishes, and room types, while expanding commonsense constraints. Ultimately, 32 concept functions were obtained, and the input of each function includes the planned itinerary elements as well as relevant information.

- **Multi-Destination Planning**. To address the limitations of the original NESY framework, which only supported single-destination planning and lacked mechanisms for sequencing multiple destinations, we implement a two-phase modification: (1) For **determining the destination sequence**, we prioritize transportation data (e.g. direct accessibility, travel duration) to determine the planning order. In the absence of such data, we default to the sequence extracted from user input via LLMs. (2) For **day allocation**, we introduce a "cyclic allocation method": each destination is initially assigned one day, with additional days distributed sequentially according to the predetermined order until meeting the total day requirement.

- **Transportation Mode Adaptation**. To address the original method's limitation of deeming planning infeasible when large-scale transportation data (e.g. flights, trains) are unavailable, we implement a more flexible approach. In the absence of such data, we now default to a "self-driving" mode. This adaptation accommodates scenarios like local trips and self-driving tours, maximizing the utilization of POI resources in areas lacking comprehensive transportation data. We establish fixed time windows for self-driving based on typical travel patterns: outbound trips are scheduled from 9:00 to 11:00, and return trips from 18:00 to 20:00.

## H    REINFORCEMENT LEARNING

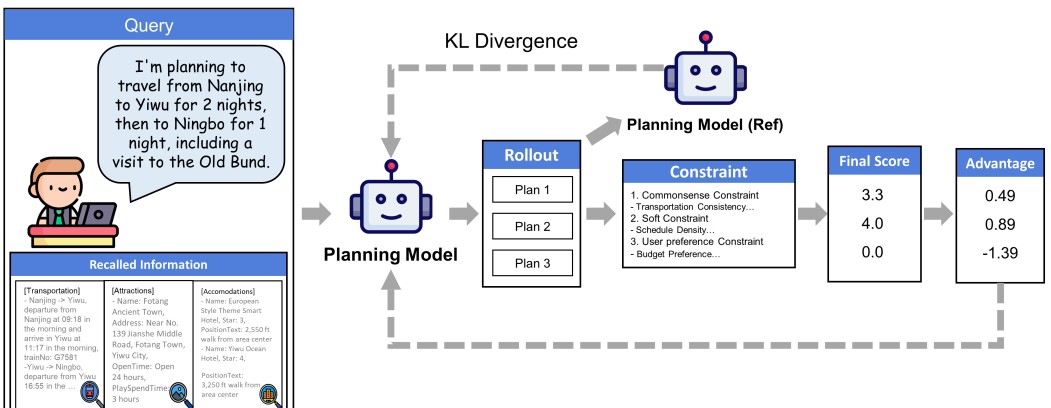

Figure 6: Demonstration of the Grouped Relative Policy Optimization (GRPO). GRPO estimates the baseline from group scores, and normalize them into a standard value. The planning model undergoes iterative optimization by maximizing the objective function $\mathcal{L}(\pi_\theta)$.

We leverage reinforcement learning to enhance the reasoning capabilities of Large Language Models (LLMs) for travel planning. We implement the Grouped Relative Policy Optimization (GRPO) (Shao et al., 2024b) algorithm, as illustrated in Figure 6. This approach is well-suited for our travel planning task due to its ability to handle fine-grained ordinal reward.

For each user query $q$, we sample a group of travel plans $\{o_1, o_2, \ldots, o_G\}$, from the current planning model $\pi_{\theta_{old}}$. These plans are then evaluated by our framework, which assigns a reward score to each plan $r = \{r_1, r_2, \ldots, r_G\}$. To improve training efficiency, we utilize only the rule-based component of our evaluation, omitting the LLM-based assessment. We normalize these rewards within each group $\tilde{r}_i = (r_i - \text{mean}(r))/\text{std}(r)$. For each plan $o_i$ in the group, we set the advantage $\hat{A}_{i,t}$, for

all tokens in the plan as the normalized reward $\hat{A}_{i,t} = \tilde{r}_i$. The planning model is then optimized by maximizing the objective $\mathcal{L}(\pi_\theta)$.

$$r_{i,t}(\theta) \coloneqq \frac{\pi_\theta(o_{i,t} \mid q, o_{i,<t})}{\pi_{\theta_{\text{old}}}(o_{i,t} \mid q, o_{i,<t})} \tag{9}$$

$$\mathcal{L}(\pi_\theta) = \frac{1}{G} \sum_{i=1}^{G} \frac{1}{|o_i|} \sum_{t=1}^{|o_i|} \left\{ \min\left[ r_{i,t}(\theta)\hat{A}_{i,t}, \ \text{clip}\left(r_{i,t}(\theta), 1-\varepsilon, 1+\varepsilon\right)\hat{A}_{i,t} \right] - \beta D_{\text{KL}}[\pi_\theta \,\|\, \pi_{\text{ref}}] \right\} \tag{10}$$

where $\beta$ serves as a coefficient that regulates the influence of the KL divergence constraint and $\varepsilon$ is a hyperparameter that determines the extent of the clipping in the surrogate objective.

# I  SOFT CONSTRAINT SCORE RULE

The soft constraints $\boldsymbol{S}_{\text{soft}}$ encompass schedule density, hotel consistency, daytime utilization, unique attractions, location clustering, iconic landmark, and attraction diversity. Formally, $\boldsymbol{S}_{\text{soft}} = (S_{\text{schedule}}, S_{\text{hotel}}, S_{\text{daytime}}, S_{\text{unique}}, S_{\text{location}}, S_{\text{iconic}}, S_{\text{diversity}})$.

**1. Schedule Density.** This metric assesses the temporal feasibility of daily itineraries by checking minimum and maximum activity hours against day-specific thresholds. Scoring is by day-level violation (at most one penalty per day):

$$S_{\text{schedule}} = 1 - \frac{|d_v|}{D} \tag{11}$$

where $D$ is the total number of days. $d_v$ is the set of days where the total activity hours exceed the upper bound or fall below the lower bound.

**2. Hotel Consistency.** The hotel consistency metric applies penalties to unnecessary hotel changes within a single city:

$$S_{\text{hotel}} = 1 - \frac{|S_{\text{switches}}|}{|H|} \tag{12}$$

where $S_{\text{switches}}$ represents the number of nights involving a switch to a nearby hotel (within 100 km) in the same city, and $|H|$ is the total number of hotel nights.

**3. Daytime Utilization.** This constraint ensures the efficient utilization of daytime hours, with a focus on avoiding idle time or overcrowded scheduling:

$$S_{\text{daytime}} = 1 - \frac{\sum_{d=1}^{D} \mathbf{1}[\text{violation}_d]}{D} \tag{13}$$

where $\text{violation}_d$ occurs when there is not arranged any activities in the morning and afternoon.

**4. Unique Attractions.** The uniqueness score employs a combination of linear and exponential penalties to quantify the degree of redundancy among selected attractions:

$$S_{\text{unique}} = \max\left( 0, 1 - \frac{|A_{\text{dup}}|}{|A_{\text{total}}|} - \sum_{a \in A_{\text{dup}}} \frac{(n_a - 1)^2 \cdot 0.05}{|A_{\text{total}}|} \right) \tag{14}$$

where $A_{\text{dup}}$ represents non-consecutively duplicated attractions. $n_a$ is the number of times an activity appears. $A_{\text{total}}$ represents the total attractions in the itinerary.

**5. Location Clustering.** This metric optimizes the spatial distribution of daily activities by applying penalties proportional to inter-attraction distances:

$$S_{\text{location}} = \max\left(0, 1 - \frac{\mathcal{P}_{\text{total}}}{|A_{\text{total}}|}\right) \tag{15}$$

where $\mathcal{P}_{\text{total}}$ represents the consecutive activity pairs for the same day that are among the top 20% farthest of all activity pairs.

**6. Iconic Landmark** and **Attraction Diversity**. These two metrics are assessed using a 5-point Likert scale, with evaluations conducted by LLMs. The resulting scores are then normalized to ensure they fall within the range of 0 to 1. The corresponding prompts are provided in Appendix K.2 and Appendix K.1. The calculation applied to both the Iconic Landmark and Attraction Diversity metrics is as follows.

$$S = \frac{\text{rating} - 1}{4}, \quad \text{rating} \in 1, 2, 3, 4, 5 \tag{16}$$

This formulation ensures that the lowest rating (1) corresponds to a normalized score of 0, while the highest rating (5) yields a normalized score of 1, providing a standardized measure for both iconic landmarks and attraction diversity.

## J    PREFERENCE CONSTRAINT SCORE RULE

**Synthetic Datasets.** The preference evaluator measures how well an itinerary aligns with the user's stated preferences. We consider four dimensions: $\boldsymbol{S}_{\text{pref}} = (S_{\text{budget}}, S_{\text{pacing}}, S_{\text{attraction}}, S_{\text{effort}})$.

1. Budget Preference ($E_{budget}$): The budget preference metric calculates the proportion of consistency between users' budget-corresponding expected hotel star ratings and actual hotel star ratings, aiming to measure the matching degree of budget and hotel star:

$$S_{\text{budget}} = \frac{1}{|H|} \sum_{h \in H} g(h; \text{pref}). \tag{17}$$

where $H$ is the set of hotel nights, and $g(h; \text{pref}) \in \{0, 1\}$ indicate whether night $h$ matches the budget preference (e.g., 0-2 stars for "cost-effective", 3-4 stars for "comfortable" and 5 stars for "High-end").

2. Pacing Preference ($E_{\text{pacing}}$): Pacing Preference quantifies the consistency between the actual itinerary arrangement and users' pace preferences based on key indicators such as play duration and the number of activities:

$$S_{\text{pacing}} = \frac{1}{D} \sum_{d=1}^{D} f(d; \text{pref}). \tag{18}$$

where $D$ is the number of days and $f(d; \text{pref}) \in [0, 1]$ is a day-level compliance function (1 when daily maximum or minimum activity time thresholds are met, intermediate value otherwise)

3. Attraction Preference ($E_{\text{attraction}}$): We map user attraction preferences to POI tags and measure coverage among visited POIs. It is calculated by the proportion of qualified attraction:

$$S_{\text{attraction}} = \frac{1}{|A_{\text{total}}|} \sum_{a \in A_{\text{total}}} m(a; \text{pref}). \tag{19}$$

where $A_{\text{total}}$ represents all visited attractions and $m(a; \text{pref}) \in \{0, 1\}$ indicating whether attraction $a$ matches any preferred tag.

4. Physical Effort Preference ($E_{\text{effort}}$): Calculates activity's physical effort value by rules, determines itinerary exertion type via high-effort activity and duration ratios, and quantifies consistency with users' physical effort preferences.

$$S_{\text{attraction}} = \frac{1}{D} \sum_{d=1}^{D} h(d; \text{pref}).$$ (20)

where $h(d; \text{pref}) \in \{0, 1\}$ indicate whether the day $d$ matches the preferred physical effort tag. A day is labeled strenuous if (i) its single-day physical-exertion score is greater than 2, where hiking, theme-park and mountain-climbing each contribute 1, cycling contributes 2, and all other activities contribute 0; or (ii) it requires physical exertion on two consecutive days. A day is light if it involves no exertion and neither adjacent day involves exertion. All remaining cases are moderate.

**Real-world Datasets.** For real-world datasets, we employ LLM-based evaluation to assess overall user request compliance. $S_{pref} = \{S_{user}\}$. The corresponding prompt is provided in Appendix K.3.

$$S_{user} = \alpha \text{LLM}(\text{prompt, response, user\_request})$$ (21)

where $\alpha$ represents the scaling weight (0.2) employed to normalize the score within the range of 0 to 1.

## K    PROMPT LIST

### K.1    ICONIC LANDMARKS EVALUATION PROMPT

---
**ICONIC_LANDMARKS_EVALUATION_PROMPT**

Please evaluate whether the attractions in the following itinerary cover the classic must-visit attractions of corresponding destination.

Itinerary: {answer_text}

Please evaluate based on the following criteria:
1. Does it include the most famous landmark attractions of the destination.
2. Does it cover different types of classic attractions (historical culture, natural scenery, modern architecture, etc.).
3. The popularity and recommendation level of the attractions.
4. Please consider the number of days in the itinerary. If some secondary attractions cannot be covered due to insufficient days, you can relax the evaluation criteria.

Please return the evaluation result in JSON format:
{
   "score": score (integer rating 1-5, where 1=no classic attractions, 2=only a few classic attractions, 3=some classic attractions, 4=most classic attractions, 5=all classic attractions),
   "missing_attractions": ["list of missing important classic attractions"],
   "explanation": "detailed explanation"
}

---

### K.2    ATTRACTION DIVERSITY EVALUATION PROMPT

---
**ATTRACTION_DIVERSITY_EVALUATION_PROMPT**

Please evaluate the richness and diversity of the following itinerary to determine if there are homogenization issues.

Itinerary: {answer_text}

Please evaluate based on the following criteria:
1. Diversity of attraction types (historical culture, natural scenery, entertainment, shopping & dining, etc.).
2. Richness of activity experiences (sightseeing, hands-on experiences, interactive, leisure, etc.).
3. Reasonable pace arrangement (balance of active/quiet, indoor/outdoor).
4. Avoiding repetitive or homogeneous activities.

---

5. Please consider the main attraction types of the destination. If the main attractions of the destination are of a single type, you can relax the evaluation criteria for homogeneity issues.

Please return the evaluation result in JSON format:
{
  "score": score (integer rating 1-5, where 1=homogenization problem accounts for more than 80% of the itinerary, 2=homogenization problem accounts for about 60% of the itinerary, 3=homogenization problem accounts for about 40% of the itinerary, 4=homogenization problem accounts for about 20% of the itinerary, 5=homogenization problem is small or nonexistent),
  "diversity_issues": ["list of identified homogenization or monotony issues"],
  "explanation": "detailed explanation"
}

## K.3 USER REQUEST FULFILLMENT EVALUATION PROMPT

### USER_PREFERENCE_CONSTRAINT_PROMPT

You are a professional travel itinerary evaluation expert, responsible for evaluating whether the generated itinerary meets the user's specific request and expectations.
Please evaluate the following travel itinerary based on the assessment criteria to determine whether it meets the user's specific request and expectations.

**User Request: **
{user_request}

**Generated Itinerary Response: **
{answer_text}

You need to carefully analyze the user's requirements and evaluate the itinerary's alignment based on the following aspects: Departure/Destination, Schedule/Timing, Mode of Transportation, Number of Travelers, Accommodation Requirements, Coverage of Attractions, Activity Types, Pace of the Trip, Budget, Other Requirements.

**Scoring Criteria**
5 points: Excellent. The itinerary fully meets all the user's requirements and considers potential personalized needs, providing a travel plan that exceeds expectations.
4 points: Good. The itinerary fully meets all the user's core requirements; however, there are details that could be further optimized.
3 points: Average. The itinerary satisfies most user requirements, such as mandatory budget, schedule, and number of travelers, but some aspects are not adequately addressed.
2 points: Poor. The itinerary fails to meet the user's main requirements, with most elements misaligned with their preferences.
1 point: Very Poor. The itinerary completely fails to meet the user's expectations and is irrelevant to their request.
0 points: The user did not provide any specific information (e.g., "Plan a trip for me"), in which case any itinerary offered can be considered as meeting the user's needs.

**Instructions for Scoring**
1. Your evaluation should focus on determining whether the provided itinerary meets the user's expectations.
2. If IDs are provided for transportation, POIs, or hotels, you may assume these details are authentic and reliable.
3. Before assigning a score, analyze the itinerary and the user's request, explaining why you assigned that score.
4. If the user's request changes midway, base your evaluation on the latest requirements.
5. You only need to evaluate the current itinerary. If the user requests multiple or alternative options, this should not result in a deduction.
6. Strictly follow the JSON format below when providing the evaluation result
Output format:
{
  "detailed_feedback": "Detailed evaluation feedback",
  "final_score": Final score (0-5)
}

## K.4 Itinerary Generation Prompt

---

**ITINERARY_GENERATION_PROMPT**

You are a travel planning expert, skilled at generating detailed travel itineraries based on user's needs and preferences, and ultimately outputting them in JSON format. Attention: User's requirements may change, you need to adapt to the latest query of the user.

[Itinerary Arrangement Rules]
1. Arrange the itinerary according to the user's requirements. Don not change the user's travel plan, including the number of days, travel cities, travel dates and etc.
2. The itinerary should be arranged in chronological order, and the time period should be divided into Morning/Afternoon/Evening.
3. A well-designed itinerary should include transportation arrangements, accommodation and key attractions, all organized in a proper chronological sequence to ensure a smooth travel experience, Pay special attention to the restrictions on the opening hours of attractions (openTimeCalendar field) to avoid scheduling visits during times when the attractions are closed or not allowing entry.
4. For transportation, prioritize choosing main stations and pay special attention to the timing of transportation arrangements, ensuring they are scheduled within intended time period. Avoid mismatches such as planning morning departures for afternoon or evening time period.
5. For accommodation, find the most suitable hotel from the most suitable hotel area. One city only need one hotel and do not change hotel in the same city.
6. The hotel should be arranged at the end of the day, and hotel arrangements should be indicated every night, except on the last day.
7. It is especially important to pay attention to the time requirements for attractions and transportation to avoid time conflicts, which could lead to an unreasonable or unachievable itinerary.
8. Ensure that the travel schedule and physical exertion are moderate, and avoid arranging too many activities in the same time period.
9. Transportation for the outbound and return trips is required, and be careful not to mix up the outbound and return trips.
10. If the user's travel duration is short (less than 2 days), it is recommended to focus on visiting the core attractions within the city.

[Reference Data Rules]
Provided reference data may include:
Attractions: poi (id: poiId, name: poiName)
Hotels: hotel (id: hotelid, name: hotelname)
Transportations: train/flight/bus/driving/ferry/ship (id: planid, name: trainNo/flightNo/shipName)
you must select hotels and transportation from reference data. But attractions are allowed to use external resources if more suitable.

[Json Format Instruction]
1. Extract exact name, type, and id when using reference data:
For attractions type: 'poi', name: poiName, id: poiId.
For hotels type: 'hotel', name: hotelname, id: hotelid;
For transportations type: 'transportation', name: trainNo/flightNo, id: planid.
2. Attractions which is not in the reference data are allowed, but must set "id": "" and when mentioning the attraction in the description, use the format **attraction**. transportation and hotel must be chosen from the reference data.
3. for the items has the same id, just output one item, do not repeat items with different name but same id.
4. the Json format is as follows:
{
  "itineraryName": "itinerary name like: 3 Days's Travel Itinerary: Shanghai to Beijing",
  "recommendReason": "the reason why this itinerary is recommended, and make user feel that this itinerary is very suitable for him/her requirements. recommend reason should be no more than 50 words",
  "dayInfos":
  [
    {
      "day": "the order of days,a integer number starting from 1",
      "scheduleTitle": "today's schedule title",
      "scheduleDetail":
      [
        {

---

"period": "the time period when the schedule begins, must choose one from Morning/Afternoon/Evening(Capitalized Initial Letter)",
    "description": "Mention all attractions/hotels/transportations using the specified markdown syntax: For attractions and hotels,you should point out names and ids, use the format `**[PoiName](poiId)**` or `**[HotelName](hotelId)**`, if the attractions is not in the reference data, use the format `**[PoiName]**`."
        "detailList":
        [
            {
              "type": "transportation/poi/hotel",
              "id": "planid/poiId/hotelid",
              "name": "trainNo/flightNo/poiName/hotelname"
            }
        ]
      }
    ]
  }
],
"tips":
  {
    "title": "tips title",
    "info": "the tips's total content should be within 50 words"
  }
}

[References]
[transportation arrangements]
{transportation_information}

[attractions reference information]
{attraction_information}

[hotels reference information]
{hotel_information}

[User Query]
{user_request}

## K.5 REACT PROMPT

### REACT_PROMPT

You are a travel planning expert, skilled at producing a detailed itinerary based on the given transportation scheme and the user's requirements and preferences.
Please complete the task by alternating between Thought, Action, and Observation steps.
The Thought phase is used to reason about the current situation. The Action phase has two types:
(1) FeasibleEnquiry [Sub Plan]: used to assess whether a sub-plan is basically feasible. You must input the sub-plan in JSON format. The sub-plan should cover a full day.
(2) Finish [Final Plan]: used to indicate task completion. You must submit the complete and final plan (in JSON format) as the argument.

[Itinerary Arrangement Rules]
1. Arrange the itinerary according to the user's requirements. Don not change the user's travel plan, including the number of days, travel cities, travel dates and etc.
2. The itinerary should be arranged in chronological order, and the time period should be divided into Morning/Afternoon/Evening.
3. A well-designed itinerary should include transportation arrangements, accommodation and key attractions, all organized in a proper chronological sequence to ensure a smooth travel experience, Pay special attention to the restrictions on the opening hours of attractions (openTimeCalendar field) to avoid scheduling visits during times when the attractions are closed or not allowing entry.
4. For transportation, prioritize choosing main stations and pay special attention to the timing of transportation arrangements, ensuring they are scheduled within intended time period. Avoid mismatches such as planning morning departures for afternoon or evening time period.

5. For accommodation, find the most suitable hotel from the most suitable hotel area. One city only need one hotel and do not change hotel in the same city.
6. The hotel should be arranged at the end of the day, and hotel arrangements should be indicated every night, except on the last day.
7. It is especially important to pay attention to the time requirements for attractions and transportation to avoid time conflicts, which could lead to an unreasonable or unachievable itinerary.
8. Ensure that the travel schedule and physical exertion are moderate, and avoid arranging too many activities in the same time period.
9. Transportation for the outbound and return trips is required, and be careful not to mix up the outbound and return trips.
10. If the user's travel duration is short (less than 2 days), it is recommended to focus on visiting the core attractions within the city.

[Reference Data Rules]
Provided reference data may include:
Attractions: poi (id: poiId, name: poiName)
Hotels: hotel (id: hotelid, name: hotelname)
Transportations: train/flight/bus/driving/ferry/ship (id: planid, name: trainNo/flightNo/shipName)
you must select hotels and transportation from reference data. But attractions are allowed to use external resources if more suitable.

[Complete Plan JSON Specification]
1. Extract exact name, type, and id when using reference data:
For attractions type: 'poi', name: poiName, id: poiId.
For hotels type: 'hotel', name: hotelname, id: hotelid;
For transportations type: 'transportation', name: trainNo/flightNo, id: planid.
2. Attractions which is not in the reference data are allowed, but must set "id": "" and when mentioning the attraction in the description, use the format **attraction**. transportation and hotel must be chosen from the reference data.
3. for the items has the same id, just output one item, do not repeat items with different name but same id.
4. the Json format is as follows:
{
  "itineraryName": "itinerary name like: 3 Days's Travel Itinerary: Shanghai to Beijing",
  "recommendReason": "the reason why this itinerary is recommended, and make user feel that this itinerary is very suitable for him/her requirements. recommend reason should be no more than 50 words",
  "dayInfos":
  [
    {
      "day": "the order of days,a integer number starting from 1",
      "scheduleTitle": "today's schedule title",
      "scheduleDetail":
      [
        {
            "period": "the time period when the schedule begins, must choose one from Morning/Afternoon/Evening(Capitalized Initial Letter)",
          "description": "Mention all attractions/hotels/transportations using the specified markdown syntax: For attractions and hotels,you should point out names and ids, use the format `**[PoiName](poiId)**` or `**[HotelName](hotelId)**`, if the attractions is not in the reference data, use the format `**[PoiName]**`."
          "detailList":
          [
            {
              "type": "transportation/poi/hotel",
              "id": "planid/poiId/hotelid",
              "name": "trainNo/flightNo/poiName/hotelname"
            }
          ]
        }
      ]
    }
  ],
  "tips":
  {
    "title": "tips title",

```
 "info": "the tips's total content should be within 50 words"
  }
}

******* Example **********
Input: I want to travel to Beijing, departing from Shanghai, for 3 days. Please design a detailed travel itinerary.
You may call FeasibleEnquiry like FeasibleEnquiry[{    "day": 1,
   "scheduleTitle": "Arrivel in Beijing and begins the tour with Beijing Zoo",
   "scheduleDetail": [
     {
      "period": "Morning",
      "description": "Take the train **T110** from Shanghai to Beijing, the train duration is approximately 6 hours",
      "detailList": [
      {
       "type": "transportation",
       "id": "1001",
       "name": "T110"
      }
      ]
     },
     {
      "period": "Afternoon",
        "description": "Exploring **[Beijing Zoo](0001)** and experience the vitality of the wildlife.",
      "detailList": [
       {
       "type": "poi",
       "id": "0001",
       "name": "Beijing Zoo"
      }
      ]
     },
     {
      "period": "Evening",
      "description": "Stay in the Tiananmen Square Area for its proximity to magor attractions. Recommend **[Beijing Tiantan Manssion Hotel](0002)** for its convenient transportation",
      "detailList": [
       {
       "type": "hotel",
       "id": "0002",
       "name": "Beijing Tiantan Manssion Hotel"
      }
      ]
     }
   ]
 }
]
You may call Finish like Finish[{
  "itineraryName": "3 Days's Travel Itinerary: Shanghai to Beijing",
  "recommendReason": "Based on your request, with just three days, traveling from Shanghai to Beijing allows you to experience both cultural landmarks and modern attractions. I recommend this itinerary to explore Beijing's highlights at a comfortable pace.",
  "dayInfos": [
    "day": 1,
    "scheduleTitle": "Arrival in Beijing and begins the tour with Beijing Zoo",
    "scheduleDetail": [
     {
      "period": "Morning",
      "description": "Take the train **T110** from Shanghai to Beijing, the train duration is approximately 6 hours",
        "detailList": [
```

```
            {
             "type": "transportation",
             "id": "1001",
             "name": "T110"
            }
           ]
          },
          {
           "period": "Afternoon",
             "description": "Exploring **[Beijing Zoo](0001)** and experience the vitality of the
        wildlife.",
           "detailList": [
            {
             "type": "poi",
             "id": "0001",
             "name": "Beijing Zoo"
            }
           ]
          },
          {
           "period": "Evening",
           "description": "Stay in the Tiananmen Square Area for its proximity to magor attractions. Recommend
        **[Beijing Tiantan Manssion Hotel](0002)** for its convenient transportation",
           "detailList": [
            {
             "type": "hotel",
             "id": "0002",
             "name": "Beijing Tiantan Manssion Hotel"
            }
           ]
          }
         ]
        },
        {
         "day": 2,
         "scheduleTitle": "Explore Universal Beijing Resort and rediscover the joys of childhood ",
         "scheduleDetail": [
          {
           "period": "Morning",
             "description": "After breakfast, take bus to **[Universal Beijing Resort](0003)**
        which is a large theme park resort located in Tongzhou District, including the Universal Studios Beijing
        theme park, two resort hotels, and a comprehensive commercial area",
           "detailList": [
            {
             "type": "poi",
             "id": "0003",
             "name": "Universal Beijing Resort"
            }
           ]
          }
         ]
        },
        {
         "day": 3,
         "scheduleTitle": "Cultural Immersion and Departure",
         "scheduleDetail": [
          {
           "period": "Morning",
           "description": "Visit **[temple of Heaven](0004)**: Explore this UNESCO Wold Heritage
        Site, where emperors prayed for good harvests",
           "detailList": [
            {
              "type": "poi",
```

```
        "id": "0004",
        "name": "temple of Heaven"
       }
      ]
     }
    ]
   }
  ],
  "tips": {
   "title": "Tips for Enhanced Travel Experience",
   "info": [
     "Use a private car or join a guided tour for the Universal Beijing Resort",
     "Book tickets in advance to secure your entry",
     "Pack comfortable walkina shoes"
   ]
  }
 }
]
```
******* End of Example **********
You must call Finish to indicate you have completed the task. Each action must call exactly one function;
do not call multiple functions at the same time. Please make sure call Finish before 5 rounds of diaglogue.

[References]
[transportation arrangements]
{transportation_information}

[attractions reference information]
{attraction_information}

[hotels reference information]
{hotel_information}

[User Query]
{user_request}

## K.6  POINT-WISE EVALUATION PROMPT

**POINT_WISE_EVALUATION_PROMPT**

You are a travel itinerary quality reviewer. Please rate a single itinerary based on the following criteria
(0-100), without introducing external information or speculation.

[Evaluation Criteria] (in order of priority from high to low)

1. Format and Facts (hard constraints, severe violations directly Inferior)

   Response structure: The output must strictly follow the requested schema. Missing or misplaced elements
   are non-compliant.
   Information verification: Transportation/hotels/attractions must come from the given text; introducing
   external facts or conjecture is treated as hallucination and deemed invalid.
   Information accuracy: Details such as names/times are consistent;
   Information relevance: Description matches corresponding attractions/events.

2. Common Sense and Feasibility (hard constraints)

   Complete information: Each destination must include necessary accommodation, essential transporta-
   tion, and key activities to ensure executability.
   Correct time sequence: Activities must be listed in temporal order; days cannot backtrack, and intra-day
   sequences must be non-decreasing in time.
   Location consistency: A traveler cannot be in multiple cities/locations simultaneously; any change of
   location must be justified by an explicit transport step.
   Feasible operating hours: Visits must occur within confirmed opening hours; closed days/times invalidate
   scheduled activities.
   Transportation block: No activities scheduled during transport intervals;

Early transportation rule: If departure time is before 10 AM, no earlier activities scheduled that day;
Transportation continuity: Smooth movement between cities/attractions, no repeated backtracking.

3. Soft Constraints

Moderate pace density: Daily pacing should be balanced—neither overpacked nor overly sparse—with reasonable buffers for transition and rest.
Hotel Consistency: Within the same city, prefer a single hotel to minimize check-in/out overhead and travel friction.
Daytime Utilization: Prioritize activities during daylight; reserve evenings for appropriate activities or rest, avoiding unproductive daytime gaps.
Unique Attractions: Avoid repeated visits to the same (or effectively identical) attractions.
Location Clustering: Group nearby attractions to reduce transit time and improve route efficiency.
Iconic Landmarks: When feasible, include representative, must-see landmarks to improve coverage and recognizability.
Attraction Diversity: Maintain variety across categories (e.g., cultural, natural, museums, landmarks) to avoid monotony.

4. Preference Matching (only considered if preferences appear in the text, otherwise treated neutrally)

Budget Preference: Select hotels/activities aligned with the stated budget profile (e.g., premium, budget-conscious, value-oriented).
Pacing Preference: Match the requested pacing (relaxed, moderate, compact) by adjusting daily activity counts and durations.
Attraction Prioritization: Prioritize categories explicitly favored by the user and ensure requested items are covered.
Physical Effort Preference: Align walking distance and intensity with the specified level (light, moderate, strenuous), managing elevation and high-exertion activities accordingly.
User Request Fulfillment: Satisfy explicit user constraints (e.g., must-visit/avoid, time windows, ordering). If no preferences are stated, no penalty or credit is applied.

[Scoring Approach]
First apply compliance deductions based on hard constraints, then provide a total score considering soft constraints and preference matching.

[Scoring Anchors]
90-100: Comprehensive, factually accurate, highly actionable, well-paced, and strongly aligned with preferences.
70-85: Largely complete, with occasional minor flaws that do not impede execution or user experience.
50-65: Moderate quality; contains several issues but remains executable.
30-45: Significant flaws (e.g., temporal/spatial conflicts, missing elements) requiring substantial revision.
0-25: Numerous severe issues or a large amount of fabricated/irrelevant information; largely unusable/inactionable.

[Output Requirement]
Strictly output 'score' (0-100), no explanations or additional text.

[User Query]
{user_request}

[Itinerary]
{itinerary}

## K.7 PAIR-WISE EVALUATION PROMPT

**PAIR_WISE_EVALUATION_PROMPT**

You are a travel itinerary quality reviewer. Your task is to compare two candidate itineraries under strict evaluation criteria, without introducing external information or speculation.

[Evaluation Criteria] (in order of priority from high to low)

1. Format and Facts (hard constraints, severe violations directly Inferior)

Response structure: The output must strictly follow the requested schema. Missing or misplaced elements are non-compliant.

Information verification: Transportation/hotels/attractions must come from the given text; introducing external facts or conjecture is treated as hallucination and deemed invalid.

Information accuracy: Details such as names/times are consistent;

Information relevance: Description matches corresponding attractions/events.

2. Common Sense and Feasibility (hard constraints)

Complete information: Each destination must include necessary accommodation, essential transportation, and key activities to ensure executability.

Correct time sequence: Activities must be listed in temporal order; days cannot backtrack, and intra-day sequences must be non-decreasing in time.

Location consistency: A traveler cannot be in multiple cities/locations simultaneously; any change of location must be justified by an explicit transport step.

Feasible operating hours: Visits must occur within confirmed opening hours; closed days/times invalidate scheduled activities.

Transportation block: No activities scheduled during transport intervals;

Early transportation rule: If departure time is before 10 AM, no earlier activities scheduled that day;

Transportation continuity: Smooth movement between cities/attractions, no repeated backtracking.

3. Soft Constraints

Moderate pace density: Daily pacing should be balanced—neither overpacked nor overly sparse—with reasonable buffers for transition and rest.

Hotel Consistency: Within the same city, prefer a single hotel to minimize check-in/out overhead and travel friction.

Daytime Utilization: Prioritize activities during daylight; reserve evenings for appropriate activities or rest, avoiding unproductive daytime gaps.

Unique Attractions: Avoid repeated visits to the same (or effectively identical) attractions.

Location Clustering: Group nearby attractions to reduce transit time and improve route efficiency.

Iconic Landmarks: When feasible, include representative, must-see landmarks to improve coverage and recognizability.

Attraction Diversity: Maintain variety across categories (e.g., cultural, natural, museums, landmarks) to avoid monotony.

4. Preference Matching (only considered if preferences appear in the text, otherwise treated neutrally)

Budget Preference: Select hotels/activities aligned with the stated budget profile (e.g., premium, budget-conscious, value-oriented).

Pacing Preference: Match the requested pacing (relaxed, moderate, compact) by adjusting daily activity counts and durations.

Attraction Prioritization: Prioritize categories explicitly favored by the user and ensure requested items are covered.

Physical Effort Preference: Align walking distance and intensity with the specified level (light, moderate, strenuous), managing elevation and high-exertion activities accordingly.

User Request Fulfillment: Satisfy explicit user constraints (e.g., must-visit/avoid, time windows, ordering). If no preferences are stated, no penalty or credit is applied.

[Decision]
First pay attention to hard constraints, severe violations are inferior; if both comply, then compare soft constraints and preference matching. If difficult to distinguish, choose the clearer and more executable one.

[Output Requirement]
Only output "Route A" or "Route B", no other characters or explanations.

[User Query]
{user_request}

[Route A]
{route_A}

[Route B]
{route_B}

