# OpenReview forum: "TripScore: Benchmarking and rewarding real-world travel planning with fine-grained evaluation"
_ICLR.cc/2026/Conference — ICLR 2026 Conference Withdrawn Submission_

### Official Review · Reviewer_vV7y · 2025-10-29

**Soundness:** 2
**Presentation:** 2
**Contribution:** 1
**Rating:** 2
**Confidence:** 5

**Summary:**

This paper proposes a new benchmark, TripScore, designed to evaluate the travel planning capabilities of large language models (LLMs). The authors also compare different methods, including standard LLMs, supervised fine-tuned LLMs, GRPO-based models, and neuro-symbolic approaches. While the topic is relevant and the implementation appears complete, the novelty and overall contribution of the paper are quite limited.

Major Comments:
1. Limited novelty of the benchmark.
As acknowledged by the authors, there already exist several travel-planning-related benchmarks and datasets. The incremental contribution of TripScore over these existing resources is unclear. The paper should clearly justify why constructing yet another benchmark is necessary and what unique aspects it provides beyond existing ones. Table 1 is not enough to demonstrate the necessary of this benchmark.
2. Lack of clarity in requirement collection.
The authors mention that task requirements were collected from human users. However, there is insufficient information about the background, diversity, and quality control of the participants. Without this information, it is difficult to assess the reliability and representativeness of the benchmark queries.
3. Unclear benchmark difficulty and comparative validation.
The difficulty level of the benchmark tasks is not analyzed or compared with existing datasets. It would strengthen the paper if the authors could show how the same methods perform on other travel-planning datasets to contextualize TripScore’s difficulty.
4. High similarity to existing work.
The paper closely resembles prior studies in both concept and presentation. For instance, the constraints discussed are similar to those in ChinaTravel. Additionally, Figure 1 in this paper appears visually and structurally similar to Figure 1 in TravelPlanners, raising concerns about originality in presentation.
5. Lack of experimental insights.
While several models are compared, the experimental section does not yield clear insights or actionable conclusions. The paper would benefit from deeper analysis—for example, discussing why certain methods perform better, and what the benchmark reveals about LLM reasoning or planning limitations.

**Strengths:**

This paper proposes a new benchmark, TripScore, designed to evaluate the travel planning capabilities of large language models (LLMs). The authors also compare different methods, including standard LLMs, supervised fine-tuned LLMs, GRPO-based models, and neuro-symbolic approaches.

**Weaknesses:**

1. Limited novelty of the benchmark.
As acknowledged by the authors, there already exist several travel-planning-related benchmarks and datasets. The incremental contribution of TripScore over these existing resources is unclear. The paper should clearly justify why constructing yet another benchmark is necessary and what unique aspects it provides beyond existing ones. Table 1 is not enough to demonstrate the necessary of this benchmark.
2. Lack of clarity in requirement collection.
The authors mention that task requirements were collected from human users. However, there is insufficient information about the background, diversity, and quality control of the participants. Without this information, it is difficult to assess the reliability and representativeness of the benchmark queries.
3. Unclear benchmark difficulty and comparative validation.
The difficulty level of the benchmark tasks is not analyzed or compared with existing datasets. It would strengthen the paper if the authors could show how the same methods perform on other travel-planning datasets to contextualize TripScore’s difficulty.
4. High similarity to existing work.
The paper closely resembles prior studies in both concept and presentation. For instance, the constraints discussed are similar to those in ChinaTravel. Additionally, Figure 1 in this paper appears visually and structurally similar to Figure 1 in TravelPlanners, raising concerns about originality in presentation.
5. Lack of experimental insights.
While several models are compared, the experimental section does not yield clear insights or actionable conclusions. The paper would benefit from deeper analysis—for example, discussing why certain methods perform better, and what the benchmark reveals about LLM reasoning or planning limitations.

**Questions:**

NA

---

> ### Author Response · Authors · 2025-11-25
> **(1/3)**
>
> We genuinely appreciate your careful reading and constructive suggestions.
>
> **W1**: Limited novelty of the benchmark.
>
> **A1**: We assert that TripScore is not merely "another benchmark," but a necessary corrective to the field's trajectory. While Table 1 summarizes features, the fundamental necessity of TripScore stems from three critical gaps in existing resources that render them insufficient for real-world agent development:
>
> 1. **The Gap in Intent Authenticity (Real vs. Synthetic)**. Existing benchmarks (e.g., TravelPlanner) primarily evaluate "Constraint Satisfaction" on synthetic templates. However, our logs reveal that **77.6%** of real users do not select the explicit constraints assumed by prior works.  We found that **39.3%** of users select only destinations and duration, and **38.3%** input vague, free-text requests. By exclusively focusing on these behaviors, TripScore prevents models from overfitting to artificial "logic puzzles", where SOTA NeSy solvers fail on our real-world track.
>
> 2. **The Gap in Evaluation Grounding (Expert-Verified vs. Unverified Heuristics)**. While recent works (e.g., TripTailor, RealTravel) attempt to assess quality using LLM-based judges, they lack ground-truth verification. This is a critical flaw. An unverified metric is an unreliable optimization target. TripScore is the first benchmark in this domain to rigorously calibrate its unified reward against human experts (achieving 60.75% agreement). This transforms the metric from a "black box" guess into a trusted proxy for human utility.
>
> 3. **The Gap in Optimization Utility (Actionable vs. Descriptive)**. Prior benchmarks offer fragmented metrics (e.g., separate pass rates and preference rankings) that are descriptive but hard to optimize. TripScore unifies feasibility and quality into a granular scalar reward. As evidenced by our GRPO results, this specific design enables End-to-End Reinforcement Learning, allowing the community to move beyond "prompt engineering" to directly fine-tuning agents for holistic planning capability.
>
> TripScore is necessary because it is the only resource that simultaneously offers behavior-grounded data, expert-verified evaluation, and an ctionable RL signal. Without it, the field risks solving artificial constraints rather than building useful agents.
>
> | Benchmark | Query Type | Query Source | Quality Evaluation | Output Form | Expert Validation |
> |-----------|------------|--------------|--------------------|-------------|-------------------|
> | TravelPlanner (Xie et al., 2024) | CB Template | Synthetic | ✗ | Pass rate | ✗ |
> | Trip Planning (Zheng et al., 2024) | CB Template | Synthetic | ✗ | Pass rate | ✗ |
> | ChinaTravel (Shao et al., 2024a) | CB Free-Text | AI-Generated |  Rule | Pass rate + Per-Dim ranking | ✗ |
> | TripTailor (Wang et al., 2025) | CB Free-Text | AI-Generated |  LLM | Pass rate + Surpass rate | ✗ |
> | RealTravel (Shao et al., 2025) | CB Free-Text | AI-Generated |  LLM | Pass rate + Preference rate | ✗ |
> | **TripScore-S (ours)** | **CB Template** | **Real logs (behav.)** | **Rule + LLM** | **Unified score** | **✓** |
> | **TripScore-R (ours)** | **Free-form req.** | **Real logs (req.)** | **Rule + LLM** | **Unified score** | **✓** |
>
> **W2**: Lack of clarity in collection
>
> **A2**: We apologize for the lack of clarity and have expanded Appendix B to detail our rigorous data collection protocol. Our dataset is derived from a production-level web application, not a small-scale experiment, ensuring both scale and authenticity.
>
> 1. **Collection Interface**. We deployed a globally accessible web application designed for effortless interaction. Users initiate plans by specifying destinations and duration, with the option to input needs via **predefined constraints** or **free-form text**. With explicit consent, we log approximately 10,000 requests per day, providing a rich stream of intrinsically motivated user intents.
>
> 2. **Quality Control & Anti-Abuse Measures**. To ensure data integrity, we implement a multi-layered quality assurance pipeline.
> - Identity Verification: Mandatory registration via email/phone to prevent bot spam.
> - Real-time Auditing: Automated monitoring to detect and ban accounts exhibiting abnormal session frequency or velocity (anti-scraping).
> - Content Safety: Strict NSFW keyword filtering and output blocking to prevent harmful content.
> - Rate Limiting: A cap of 100 interactions per user/day to discourage abuse while allowing normal usage.
> This robust infrastructure ensures that TripScore captures high-quality, authentic planning requirements that reflect real-world diversity.

---

> > ### Author Response · Authors · 2025-11-25
> > **(2/3)**
> >
> > **W3**: Unclear benchmark difficulty and comparative validation. The difficulty level of the benchmark tasks is not analyzed or compared with existing datasets. It would strengthen the paper if the authors could show how the same methods perform on other travel-planning datasets to contextualize TripScore’s difficulty.
> >
> > **A3**: We clarify that raw "Pass Rate" is a misleading proxy for difficulty when comparing fundamentally different paradigms. TripScore introduces the dimensions of plan quality and user request understanding, in our benchmark.
> >
> > 1. Redefining Difficulty: Logic vs. Ambiguity. In TravelPlanner, difficulty is scaled by increasing rigid constraints, leading to low pass rates for LLMs (4.4%) but allowing NeSy solvers to "game" the system. In TripScore, difficulty lies in handling real-world utility. This is proven by the catastrophic failure of NeSy solvers: while they thrive on other benchmarks (e.g., 31.6% in ChinaTravel), they achieve a near-zero Delivery Rate (1.82%) on our real-world track. This proves that TripScore is exceptionally difficult for rigid solvers, exposing a critical blind spot in current research.
> >
> > 2. High Pass Rate $\neq$ Solved Problem. While direct prompting GPT-4o achieves a 72.38% pass rate on TripScore, this only indicates that the plan is feasible, not that it is optimal. A key contribution of TripScore is shifting focus from binary success to Plan Quality. Even with a high pass rate, we introduce plan quality beyond feasibility, enabling direct comparison of plan quality.
> >
> > TripScore is not "easier". It is qualitatively different. It penalizes the artificial rigidity rewarded by prior benchmarks and challenges models to optimize utility, which is a harder, more realistic goal than merely satisfying a logic constraints checklist.
> >
> >
> > **W4**: High similarity to existing work. The paper closely resembles prior studies in both concept and presentation. For instance, the constraints discussed are similar to those in ChinaTravel. Additionally, Figure 1 in this paper appears visually and structurally similar to Figure 1 in TravelPlanners, raising concerns about originality in presentation.
> >
> > **A4**: We take the concern regarding originality seriously and have made significant revisions to clarify our distinctive contributions.
> >
> > 1. **Visual Originality** (Figure 1). We acknowledge the visual resemblance noted by the reviewer. To resolve this, we have completely redesigned Figure 1 as a distinct, arrow-based workflow diagram that better illustrates our unique "Unified Reward" pipeline, visually distancing it from the structural layouts of prior works.
> >
> > 2. **Constraints Difference**. To clarify the constraints difference, we mark our newly proposed constraints in blue in Table 4. Furthermore, we add Constraint-level comparison with existing benchmarks in the Appendix F.
> >
> > ✓ = evaluated; ✥= similar constraint exists; ✗ = not evaluated.
> > | Dimension | Ours (Eval) | ChinaTravel (Eval) | TravelPlanner (Eval) | TripTailor (Eval) |
> > |-----------|---------------|--------------------|----------------------|-------------------|
> > | **Format** | |  | | |
> > | Response format | ✓(Rule) | ✓(Rule) | ✓(Rule) | ✓(Rule) |
> > | Info verification |  ✓(Rule) | ✓(Rule) | ✓(Rule) | ✓(Rule) |
> > | Info accuracy | ✓(Rule) | ✓(Rule) | ✓(Rule) | ✓(Rule) |
> > | Info relevance | ✓(Rule) | ✗ | ✗ | ✗ |
> > | **Commonsense**|  | | | | | |
> > | Info completeness | ✓(Rule) | ✓(Rule) | ✓(Rule) | ✓(Rule) |
> > | Chronological order | ✓(Rule) | ✓(Rule) | ✗ | ✗ |
> > | Location consistency | ✓(Rule) | ✓(Rule) | ✓(Rule) | ✓(Rule) |
> > | Operating hours | ✓(Rule) | ✓(Rule) | ✗ | ✗ |
> > | Travel block-out | ✓(Rule) | ✗ | ✗ | ✗ |
> > | Transport consistency | ✓(Rule) | ✓(Rule) | ✓(Rule) | ✓(Rule) |
> > | **Soft** | | | | | |
> > | Schedule density | ✓(Rule) | ✗ | ✗ | ✗ |
> > | Hotel consistency | ✓(Rule) | ✗ | ✗ | ✗ |
> > | Daytime utilization | ✓(Rule) | ✗ | ✗ | ✗ |
> > | Unique attractions | ✓(Rule) | ✥(Rule) | ✗ | ✥(Rule) |
> > | Location clustering | ✓(Rule) | ✥(Rule) | ✗ | ✥(Rule) |
> > | Iconic landmarks coverage | ✓(LLM) | ✗ | ✗ | ✗ |
> > | Attraction diversity | ✓(LLM) | ✗ | ✗ | ✗ |
> > | **Preference** | | | | | |
> > | Budget style | ✓(Rule) | ✥(Rule) | ✥(Rule) | ✥(Rule) |
> > | Pacing style | ✓(Rule) | ✗ | ✗ | ✗ |
> > | Attraction prioritization | ✓(Rule) | ✗ | ✗ | ✗ |
> > | Physical effort | ✓(Rule) | ✗ | ✗ | ✗ |
> > | User request | ✓(LLM) | ✗ | ✗ | ✓(LLM) |

---

> > > ### Author Response · Authors · 2025-11-25
> > > **(3/3)**
> > >
> > > **W5**: Lack of experimental insights
> > >
> > > **A5**: We sincerely thank the reviewer for pointing this out. We realized that our previous draft focused too much on the "what" and not enough on the "why." In the revision, we have completely restructured the Experimental Analysis section to center around four key insights derived from our benchmark results.
> > >
> > > 1. **The efficiency-performance trade-off in Test-Time compute**. While test-time methods (e.g., LLM-Modulo, HyperTree) generally improve plan feasibility, they incur a prohibitive latency cost. As shown in Table 2, LLM-Modulo with GPT-4o achieves the highest reward (2.69) and Delivery Rate (90.86%) in real-world settings, but this comes at the expense of tripling the inference time compared to Direct prompting (62.39s vs. 21.36s). This observation suggests distinct diminishing returns for inference-time compute in planning tasks. While iterative refinement is effective, the latency overhead scales poorly for real-time applications. The marginal gain in reward often fails to justify the linear or exponential increase in compute time.
> > >
> > > 2. **The rigidity of Neuro-Symbolic solvers**. Neuro-symbolic approaches (TTG, NESY) demonstrate a over-constrained'' states where no feasible solution exists (resulting in execution failure). Constraint Paradox'', as they achieve near-perfect Constraint Pass Rates (99%) but suffer from exceptionally low Delivery Rates (DR) and Rewards. This result highlights the mismatch between hard-constraint solvers and real-world ambiguity. Symbolic solvers treat travel constraints as rigid logical rules, leading to "over-constrained'' states where no feasible solution exists (resulting in execution failure). In contrast, end-to-end LLMs perform "soft'' constraint satisfaction, finding viable trade-offs that, while not satisfying all constraints,  yield executable plans (higher DR).
> > >
> > > 3. **For small models, test-time methods are ineffective**. A striking disparity exists between model sizes when utilizing test-time compute methods. On Qwen3-8B, advanced test-time compute methods like HyperTree not only fail to outperform but also markedly under-perform Direct prompting (Reward drops from -1.00 to -2.62). We believe that smaller models (e.g., 8B parameters) lack robust instruction-following and self-verification capabilities, making it hard to guide the planning process. Thus, providing additional test-time compute to capacity-limited models introduces noise rather than refining the final solution. Effective planning process appears to be an emergent ability that requires a stronger base reasoner such as GPT-4o.
> > >
> > > 4. **RL fine-tuning internalizes reasoning. Fine-tuning methods, particularly GRPO, offer the best trade-off between plan quality and inference speed**. On Qwen3-8B, GRPO achieves a Real-world Reward of 2.04, significantly outperforming Direct prompting (-1.00) and surpassing the much larger GPT-4o Direct baseline (1.61), while maintaining a latency of under 10 seconds. This validates the hypothesis that complex planning constraints can be "internalizes" as parametric knowledge. This suggests that for domain-specific planning, parameter adaptation is far more compute-efficient than test-time search, bridging the gap between small open-source models and leading proprietary models.
> > >
> > > Furthermore, to disentangle "Plan Quality" from "Feasibility", we conducted a new **pairwise comparison experiment**. One might question whether the higher rewards of fine-tuned models are simply due to fewer constraint violations (higher CPR). To investigate this, we controlled for feasibility by only comparing instances where both methods generated feasible plans.
> > >
> > > | Test Set Pair (A vs B) | A Win Rate | 95% CI | p-value |
> > > |------------------------|------------|--------|---------|
> > > | **TripScore Reward – Real-world** | | | |
> > > | Direct (GPT-4o) vs Direct (Qwen3-14B) | 0.744 | [0.589, 0.854] | 0.004 |
> > > | Direct (GPT-4o) vs SFT (Qwen3-14B) | 0.659 | [0.505, 0.784] | 0.032 |
> > > | Direct (GPT-4o) vs RFT (Qwen3-14B) | 0.552 | [0.375, 0.716] | 0.711 |
> > > | Direct (GPT-4o) vs GRPO (Qwen3-14B) | 0.522 | [0.486, 0.557] | 0.245 |
> > > | **Expert Selection – Real-world** | | | |
> > > | Direct (GPT-4o) vs Direct (Qwen3-14B) | 0.785 | [0.671, 0.829] | 0.002 |
> > > | Direct (GPT-4o) vs SFT (Qwen3-14B) | 0.614 | [0.572, 0.654] | 0.008 |
> > > | Direct (GPT-4o) vs RFT (Qwen3-14B) | 0.561 | [0.409, 0.682] | 0.636 |
> > > | Direct (GPT-4o) vs GRPO (Qwen3-14B) | 0.513 | [0.478, 0.548] | 0.492 |
> > >
> > > As shown in the above table, while GPT-4o dominates the base Qwen3-14B (78.5% expert preference), **GRPO-tuning effectively closes this quality gap** (GPT-4o preference drops to 51.3%, statistically tied). This confirms that RL fine-tuning (GRPO) does more than just teaching the model to follow hard rules; it effectively aligns the model's latent planning process with human preferences, enabling a 14B model to generate travel itineraries that experts find qualitatively indistinguishable from those of GPT-4o.

---

> > > > ### Comment · Reviewer_vV7y · 2025-11-26
> > > >
> > > > Thanks for the author's response. I appreciate the author's efforts and clarity, so I tend to raise the score.

---

### Official Review · Reviewer_339T · 2025-10-30

**Soundness:** 2
**Presentation:** 2
**Contribution:** 1
**Rating:** 2
**Confidence:** 4

**Summary:**

This paper introduces a comprehensive benchmark designed to evaluate and improve the complex travel-planning capabilities of LLMs.
It adopts real-world user queries and constructs a comprehensive evaluation framework.
The extensive experiments show that the designed reward is beneficial in RL training.

**Strengths:**

1. This paper adopts real-world user queries, bridging the gap between previous benchmarks and applications.
2. It proposes a unified and actionable reward score, which can be further used as  a reward model for such tasks and provides deeper insights beyond pass/fail.
3. This paper conducts extensive experiments on different methods and base models.

**Weaknesses:**

1. Although the authors claim the dataset to be a real-world dataset, it only contains 219 real-world queries out of 4870 total queries, which is not convincing enough for me to view this dataset as a real-world dataset.
2. The evaluation contains LLM-as-judge and the results in Table 6 show that only 61.32% is correctly evaluated, which poses doubt on the precision of such method.
3. This paper is an incremental extension of TravelPlanner by extending constraints and constructing a trivial evaluation method.

**Questions:**

1. How is TripScore's unified reward a fundamental advance over these existing, complex reward and evaluation models?
2. The TripScore evaluator relies on an LLM to score soft and preference constraints. The authors admit this model only achieves "moderate agreement (60.75%)" with human travel experts. This implies that for nearly 40% of comparisons, the benchmark's ground truth may be wrong. Why should this 60.75%-accurate reward signal be trusted to train a superior model, and how does this level of noise affect the stability and reliability of the GRPO training?
3. What is the fundamental difference between TripScore and existing benchmarks?

---

> ### Author Response · Authors · 2025-11-25
> **(1/3)**
>
> Thank you for your insightful comments and valuable suggestions.
>
> **W1**: It only contains 219 real-world queries out of 4870 total queries, which is not convincing as a real-world dataset.
>
> **A1**: We clarify that the 219 queries in the TripScore-R (Real-world) track are strictly reserved for evaluation (Test Set), not training. For a test set intended to measure generalization, 219 diverse, authentic samples provide sufficient statistical power to distinguish model performance.
>
> Furthermore, the entire benchmark, including the larger synthetic track, is grounded in real-world behavior, not arbitrary templates.
>
> 1. **Behavior-Driven Design**. Our dataset distribution mirrors actual user logs:
>
>   - **TripScore-S (Synthetic Track)**. Models the **39.3%** of users who only provide minimal constraints (Destination + Duration). This is not "fake" data; it faithfully represents the largest segment of real-world usage.
>
>   - **TripScore-R (Real-world Track)**: Captures **38.3%**  of users with free-form, ambiguous requests.
>
> 2.  **High-Value Scarcity**. Authentic free-form planning requests from intrinsically motivated users (not crowd-workers) are extremely scarce and high-value. Unlike crowd-sourced data which can be scaled cheaply but lacks depth, each of these 219 queries represents a genuine, complex user intent that serves as a "gold standard" for testing robustness.
>
> In summary, while the raw count of TripScore-R is smaller, the  **entirety** of TripScore is behaviorally aligned with reality, and the Real-world track serves as a rigorous, high-quality probe for generalization.
>
> **W2&Q2**: The evaluation contains LLM-as-judge and the results in Table 6 show that only 61.32% is correctly evaluated, which poses doubt on the precision of such method.
>
>  **A2**: We acknowledge the reviewer's concern regarding the 60.15% accuracy of our evaluator. To address this, we have conducted a rigorous **human expert evaluation**. We paired the outputs of our trained models (Qwen3-14B) against the leading proprietary model (GPT-4o) and asked human experts to blindly select the superior plan. We also report the preference of TripScore. The results show that while individual TripScore rewards may have variance, TripScore is a valid proxy for human preference at the population level.
>
> 1. **Consistency between Reward and Human Experts**. We compared the preference rates derived from our automated TripScore Reward against blind Expert Selection on the same set of query pairs. As shown in Table 2, the trends are remarkably consistent. For instance, in the Real-world dataset, both metrics show a statistically significant gap between GPT-4o and Direct Qwen3-14B (Expert: 78.5% win rate; TripScore: 74.4%), and both metrics show this gap vanishing with GRPO (Expert: 51.3%; TripScore: 52.2%). This alignment confirms that despite potential noise at the instance level, our reward metric accurately reflects aggregate model quality and human preference trends.
>
> 2. **RL Robustness to Noisy Signals**. Crucially, the results demonstrate that our RL fine-tuning (GRPO) is robust to the imperfections of the reward signal. Even if the reward model is not 100% precise, it provides a sufficiently directional signal to guide optimization. The expert evaluation confirms that GRPO successfully translates this "noisy" reward into genuine quality improvements, enabling the 14B model to tie with GPT-4o in human evaluation, significantly outperforming the SFT baseline (61.4% GPT-4o win rate).
>
> | Test Set Pair (A vs B) | A Win Rate | 95% CI | p-value |
> |------------------------|------------|--------|---------|
> | **TripScore Reward – Synthetic** | | | |
> | Direct (GPT-4o) vs Direct (Qwen3-14B) | 0.726 | [0.614, 0.845] | 0.002 |
> | Direct (GPT-4o) vs SFT (Qwen3-14B) | 0.587 | [0.421, 0.686] | 0.632 |
> | Direct (GPT-4o) vs RFT (Qwen3-14B) | 0.515 | [0.352, 0.675] | 1.000 |
> | Direct (GPT-4o) vs GRPO (Qwen3-14B) | 0.459 | [0.284, 0.639] | 0.484 |
> | **TripScore Reward – Real-world** | | | |
> | Direct (GPT-4o) vs Direct (Qwen3-14B) | 0.744 | [0.589, 0.854] | 0.004 |
> | Direct (GPT-4o) vs SFT (Qwen3-14B) | 0.659 | [0.505, 0.784] | 0.032 |
> | Direct (GPT-4o) vs RFT (Qwen3-14B) | 0.552 | [0.375, 0.716] | 0.711 |
> | Direct (GPT-4o) vs GRPO (Qwen3-14B) | 0.522 | [0.486, 0.557] | 0.245 |
> | **Expert Selection – Synthetic** | | | |
> | Direct (GPT-4o) vs Direct (Qwen3-14B) | 0.701 | [0.604, 0.814] | 0.004 |
> | Direct (GPT-4o) vs SFT (Qwen3-14B) | 0.549 | [0.479, 0.586] | 0.137 |
> | Direct (GPT-4o) vs RFT (Qwen3-14B) | 0.504 | [0.418, 0.581] | 0.839 |
> | Direct (GPT-4o) vs GRPO (Qwen3-14B) | 0.477 | [0.365, 0.603] | 0.628 |
> | **Expert Selection – Real-world** | | | |
> | Direct (GPT-4o) vs Direct (Qwen3-14B) | 0.785 | [0.671, 0.829] | 0.002 |
> | Direct (GPT-4o) vs SFT (Qwen3-14B) | 0.614 | [0.572, 0.654] | 0.008 |
> | Direct (GPT-4o) vs RFT (Qwen3-14B) | 0.561 | [0.409, 0.682] | 0.636 |
> | Direct (GPT-4o) vs GRPO (Qwen3-14B) | 0.513 | [0.478, 0.548] | 0.492 |

---

> ### Author Response · Authors · 2025-11-25
> **(2/3)**
>
> **W3&Q3**: This paper is an incremental extension of TravelPlanner by extending constraints and constructing a trivial evaluation method. What is the fundamental difference between TripScore and existing benchmarks?
>
> **A3**:  We respectfully disagree that TripScore is merely an incremental extension. While both involve travel planning, TripScore represents a fundamental paradigm shift from "Constraint-Satisfaction" to "Preference-Optimization", driven by evidence that existing benchmarks (like TravelPlanner) are misaligned with real-world needs.
>
> 1. **Divergence in Core Objective: Logic Puzzles vs. Ambiguity Resolution**. TravelPlanner evaluates the ability to solve rigid "logic puzzles" (e.g., "visit X, budget Y") and use pass rate as the metrics. This design unintentionally favors Neuro-Symbolic (NeSy) solvers, since constraint solving can achieve very high pass rates. TripScore exposes the fragility of this approach. Our experiments reveal a critical generalization gap. NeSy methods (TTG/NESY) that dominate TravelPlanner and ChinaTravel fail catastrophically on TripScore's real-world track (suffering from low Delivery Rates). This proves that TripScore reveals the ambiguity and trade-offs among constraints, which is the true bottleneck for real-world travel planning task.
>
> 2. **Data Provenance: Artificial Templates vs. Organic Intent**. TravelPlanner or existing benchmark rely on synthetic or AI-generated templates that assume users will supply explicit, structured constraints. TripScore is grounded on authentic user behaviour. All our data come from a custom travel-planning web application where every request originates from real-world needs. We find that real users rarely select the explicit constraint checklists assumed by prior work (only 22.4\% do). By capturing the 77\% of users with vague requests or minimal constraints (only destinations and duration), TripScore evaluates a planner’s ability to generate high-quality travel plans rather than merely feasible ones.
>
> 3. **Evaluation as an RL Enabler: Binary Pass vs. Unified Reward**. TravelPlanner provides a binary Pass/Fail metric, which is sparse and single-dimensional. TripScore introduces a granular, Human-Validated Unified Reward. Unlike prior works that treat quality as a secondary ranking metric, we fuse feasibility and quality into a single scalar signal. Crucially, this transforms the benchmark from a static testbed into a dynamic Optimization Objective. As shown in our GRPO results, this granular signal enables End-to-End Reinforcement Learning to directly optimize plan utility, which is a methodological leap impossible with TravelPlanner's binary metrics.

---

> > ### Author Response · Authors · 2025-11-25
> > **(3/3)**
> >
> > **Q1**: How is TripScore's unified reward a fundamental advance over these existing, complex reward and evaluation models?
> >
> > **A4**: TripScore's unified reward represents a fundamental advance because it shifts the metric from being merely a static ruler to a differentiable optimization objective.
> >
> > 1. **From Discrete Verification to Gradient-Based Optimization (RL Enabler)**. Existing evaluation models are often fragmented (separate pass rates vs. preference rankings), making them unsuitable as dense reward signals for training. The fundamental advance of TripScore is its actionability. By fusing feasibility constraints and nuanced quality preferences into a single, granular scalar signal, TripScore transforms travel planning from a discrete search problem into a learnable Reinforcement Learning task. Our experiments empirically prove that this specific reward signal enables End-to-End RL (e.g., GRPO) to significantly improve performance, allowing open-source models to internalize complex reasoning and close the gap with proprietary models like GPT-4o.
> >
> > 2. **Grounding in Human Utility vs. Unverified Heuristics**. Existing benchmarks (e.g., in TripTailor or RealTravel) rely on unverified heuristics or "LLM-as-a-Judge" prompting. TripScore is the only benchmark in this domain rigorously calibrated against human experts. Our unified score achieves 60.75% agreement with travel experts, which significantly outperforms standard LLM-as-a-Judge baselines. This ensures that when models optimize for TripScore, they are actually aligning with human-verified standards of quality, not just overfitting to an arbitrary heuristic.
> >
> > | Benchmark | Query Type | Query Source | Quality Evaluation | Output Form | Expert Validation |
> > |-----------|------------|--------------|--------------------|-------------|-------------------|
> > | TravelPlanner (Xie et al., 2024) | CB Template | Synthetic | ✗ | Pass rate | ✗ |
> > | Trip Planning (Zheng et al., 2024) | CB Template | Synthetic | ✗ | Pass rate | ✗ |
> > | ChinaTravel (Shao et al., 2024a) | CB Free-Text | AI-Generated | Rule | Pass rate + Per-Dim ranking | ✗ |
> > | TripTailor (Wang et al., 2025) | CB Free-Text | AI-Generated | LLM | Pass rate + Surpass rate | ✗ |
> > | RealTravel (Shao et al., 2025) | CB Free-Text | AI-Generated | LLM | Pass rate + Preference rate | ✗ |
> > | **TripScore-S (ours)** | **CB Template** | **Real logs (behav.)** | **Rule + LLM** | **Unified score** | **✓** |
> > | **TripScore-R (ours)** | **Free-form req.** | **Real logs (req.)** | **Rule + LLM** | **Unified score** | **✓** |

---

### Official Review · Reviewer_h4xw · 2025-11-01

**Soundness:** 3
**Presentation:** 3
**Contribution:** 3
**Rating:** 6
**Confidence:** 3

**Summary:**

This work addresses a critical gap in travel planning benchmarks by unifying fine-grained evaluation criteria into a single reward and incorporating real-world user queries. The 4,870-query dataset (including 219 real-world requests) and four-category constraint framework (format, commonsense, soft, preference) are valuable contributions, filling the void of authentic scenario coverage in existing benchmarks.

**Strengths:**

1. Addresses a key limitation of existing travel planning benchmarks (e.g., TravelPlanner, ChinaTravel) by unifying four types of fine-grained constraints into a single interpretable reward score, providing a more coherent and scalable evaluation mechanism.

2. Constructs a high-quality real-world dataset, mitigating the overreliance on LLM-generated data in prior benchmarks and enhancing generalization and practical applicability.

3. Extensive experiments with diverse algorithms demonstrate the effectiveness of reinforcement learning for travel plan generation.

**Weaknesses:**

1. The framework relies on LLM-based soft or preference evaluation, which introduces potential inaccuracy and computational overhead due to the absence of a purely rule-based alternative.

2. Given the proliferation of travel planning benchmarks, including ChinaTravel, the novelty appears somewhat limited in terms of benchmark design.

**Questions:**

Q1: Why do strict hard constraints often lead to no feasible solutions in NeSy-based approaches to travel plan generation?

Q2: What are the benefits of using a unified evaluation metric in this context?

Q3: Compared with ChinaTravel, which focuses on NeSy reasoning, what type of approach do you personally find more promising for advancing travel planning tasks, and what are your key insights?

Q4: Why was the ReAct baseline not included in the experiments?

---

> ### Author Response · Authors · 2025-11-25
> **(1/4)**
>
> We truly appreciate your professional review and invaluable feedback, which helps improve our paper.
>
> **W1**: The framework relies on LLM-based soft or preference evaluation, which introduces potential inaccuracy and computational overhead.
>
> **A1**:  1. During RL fine-tuning (e.g., GRPO), we intentionally **disable the LLM-based evaluation components** and rely solely on the rule-based reward (the constraints are evaluated in Rule in Table 4). This strategy eliminates the computational overhead during the millions of inference steps required for policy optimization, ensuring scalability without sacrificing the richness of the final evaluation.
>
> 2. To rigorously confirm that rule-based training also generalizes to human-preferred quality, we conducted a comprehensive **pairwise human expert evaluation** (Table 3). We benchmarked the preference rates derived from our automated TripScore Reward (LLM-enabled) against blind Expert Selection on the same set of query pairs. As demonstrated in the table below, the results prove that our RL-tuned model, despite being trained on rule-based rewards, successfully generalizes to human-expert quality standards. Under blind expert review, GRPO reduces the preference for GPT-4o from a dominant 78.5% (vs. Base) to a statistical tie of 51.3%, effectively matching the proprietary model's quality.
>
> | Test Set Pair (A vs B) | A Win Rate | 95% CI | p-value |
> |------------------------|------------|--------|---------|
> | **TripScore Reward - Synthetic** | | | |
> | Direct (GPT-4o) vs Direct (Qwen3-14B) | 0.726 | [0.614, 0.845] | 0.002 |
> | Direct (GPT-4o) vs SFT (Qwen3-14B) | 0.587 | [0.421, 0.686] | 0.632 |
> | Direct (GPT-4o) vs RFT (Qwen3-14B) | 0.515 | [0.352, 0.675] | 1.000 |
> | Direct (GPT-4o) vs GRPO (Qwen3-14B) | 0.459 | [0.284, 0.639] | 0.484 |
> | **TripScore Reward - Real-world** | | | |
> | Direct (GPT-4o) vs Direct (Qwen3-14B) | 0.744 | [0.589, 0.854] | 0.004 |
> | Direct (GPT-4o) vs SFT (Qwen3-14B) | 0.659 | [0.505, 0.784] | 0.032 |
> | Direct (GPT-4o) vs RFT (Qwen3-14B) | 0.552 | [0.375, 0.716] | 0.711 |
> | Direct (GPT-4o) vs GRPO (Qwen3-14B) | 0.522 | [0.486, 0.557] | 0.245 |
> | **Expert Selection - Synthetic** | | | |
> | Direct (GPT-4o) vs Direct (Qwen3-14B) | 0.701 | [0.604, 0.814] | 0.004 |
> | Direct (GPT-4o) vs SFT (Qwen3-14B) | 0.549 | [0.479, 0.586] | 0.137 |
> | Direct (GPT-4o) vs RFT (Qwen3-14B) | 0.504 | [0.418, 0.581] | 0.839 |
> | Direct (GPT-4o) vs GRPO (Qwen3-14B) | 0.477 | [0.365, 0.603] | 0.628 |
> | **Expert Selection - Real-world** | | | |
> | Direct (GPT-4o) vs Direct (Qwen3-14B) | 0.785 | [0.671, 0.829] | 0.002 |
> | Direct (GPT-4o) vs SFT (Qwen3-14B) | 0.614 | [0.572, 0.654] | 0.008 |
> | Direct (GPT-4o) vs RFT (Qwen3-14B) | 0.561 | [0.409, 0.682] | 0.636 |
> | Direct (GPT-4o) vs GRPO (Qwen3-14B) | 0.513 | [0.478, 0.548] | 0.492 |
>
> **W2**: The novelty appears somewhat limited in terms of benchmark design.
>
> **A2**: We thank the reviewer for the opportunity to clarify our unique positioning. TripScore introduces fundamental advancements that address critical blind spots in current evaluation paradigms.
>
> 1. **Exposing the Generalization Gap**. We identify a major flaw in existing constraint-based benchmarks (e.g., TravelPlanner). They unintentionally favor neuro-symbolic solvers (TTG, NESY) that excel at rigid logic puzzles but fail in reality. Our experiments reveal a stark contrast. While these solvers achieve near-perfect scores on given constraints, they suffer catastrophic failures on our real-world queries (low delievery rate and low Reward). TripScore is the first to quantitatively demonstrate this generalization gap, proving that "solving constraints" $\neq$ "planning well."
>
> 2. **Real Intent vs. AI-generated**. Benchmarks like ChinaTravel and TripTailor rely heavily on AI-generated queries or rigid templates. In contrast, TripScore is grounded in real-world user logs, revealing that users rarely choose the explicit constraints assumed by prior works.  As our logs reveal, 39.3% of users select only destinations and duration, and 38.3% input vague, free-text requests. Only 22.4% of users will select given constraints checklist. By shifting focus to the 77% of minimal constaints and vague input, TripScore evaluates the planner's ability to infer utility, not just pass the logical constraints.
>
> 3. Unlike the discrete "Pass Rates" or fragmented metrics of prior works, TripScore proposes a Unified Reward mechanism that fuses feasibility and quality into a granular scalar signal. This is not just an evaluation metric but an **RL enabler**. As evidenced by our GRPO results, this granular signal allows us to fine-tune models to match proprietary performance, bridging the gap in a way that static benchmarks cannot.
>
> 4. Most critically, TripScore is the benchmark backed by rigorous **human expert validation**. While ChinaTravel and RealTravel rely on unverified rules or LLM-judges, our metric is calibrated against human experts, ensuring it reflects true real-world utility.

---

> ### Author Response · Authors · 2025-11-25
> **(2/4)**
>
> **Q1**: Why do strict hard constraints often lead to no feasible solutions in NeSy-based approaches to travel plan generation?
>
> **A3**: The high failure rate of NeSy-based approaches stems from a fundamental mismatch between rigid symbolic logic and the ambiguous, flexible nature of real-world travel planning. While NeSy systems excel at verifiable logic, they often lack the semantic flexibility to handle "soft" constraints, translating vague user intents into overly strict, unexecutable logical predicates.
>
> We illustrate this with three specific failure modes observed in our experiments:
>
> 1. **Semantic Misinterpretation: Treating Abstract Intents as Concrete Entities**. Users frequently request activities that are not discrete physical locations (POIs) in a database, such as "attend a meeting" or "rest." NeSy translators, lacking semantic grounding, often erroneously map these abstract concepts to rigid POI-visit constraints.
>
>  Example Query: "Trip around Beijing. I have a meeting all day tomorrow and want to rest on Thursday"
>
>  Generated DSL Constraint:
>
>     ```python
>     # The solver interprets "Meeting" and "Rest" as specific POI names to be visited
>     result_day2 = (day2_attraction_names_set <= {'Meeting'})
>     result_day3 = (day3_attraction_names_set <= {'Rest'})
>     ```
>
>  Failure Cause: The solver attempts to find attractions literally named "Meeting" or "Rest" within the candidate set. Since no such entities exist, the solver returns "Unsatisfiable." In contrast, an end-to-end LLM would correctly interpret this as a directive to leave a time slot empty or instantiate a custom event outside the provided candidate set.
>
> 2. **Rigid Entity Matching: The "Exact Match" Trap**. Symbolic solvers typically enforce strict set membership or equality checks. If a user provides a generic term (e.g., "Jiuzhaigou hotels") while the database uses specific official names, the rigid constraint fails to bridge the semantic gap.
>
>  Example Query: "Jiuzhaigou 3-day trip. Jiuzhaigou hotels and attractions."
>
> Generated DSL Constraint:
>
>     ```python
>     # The solver requires the selected hotel's name to be strictly "Jiuzhaigou"
>     result = ({'Jiuzhaigou'} <= hotel_names_set)
>     ```
>
>  Failure Cause: Real-world database entries are specific (e.g., *"InterContinental Resort Jiuzhaigou Paradise"*). The strict logical condition `{'Jiuzhaigou'} <= {'InterContinental...'}` evaluates to `False`. The solver lacks the flexibility to perform semantic fuzzy matching or understand that the user implies "any hotel located in Jiuzhaigou."
>
> 3. **Oversimplified Logic Generation**.The Common-Sense Gap. NeSy translators often struggle to convert complex, implicit constraints (like "overnight train") into accurate logical predicates, resorting to brittle heuristics that fail to capture the true condition.
>
>  Example Query: "Trip starting from Guangzhou, March 27th-31st. I want to spend the night on the train."
>
>  Generated DSL Constraint:
>
>     ```python
>     # The solver simplifies "overnight on train" to a naive time check
>     overnight_train_found = False
>     if departure_time >= '23:00' or arrival_time <= '06:00':
>         overnight_train_found = True
>     result = overnight_train_found
>     ```
>
>  Failure Cause: The generated logic is logically flawed or too restrictive. It naively assumes "overnight" means departing after 11 PM or arriving before 6 AM. It misses standard overnight trains that might depart at 8 PM and arrive at 8 AM the next day. Consequently, valid overnight options are filtered out, leading to unnecessary plan failures.
>
> These examples highlight that hard constraints demand a level of precision in prompt-to-logic translation that is inherently brittle in open-domain scenarios. NeSy solvers are blocked by their own logical strictness when facing ambiguity, whereas LLMs can employ soft reasoning to resolve these gaps.

---

> ### Author Response · Authors · 2025-11-25
> **(3/4)**
>
> **Q2**: What are the benefits of using a unified evaluation metric in this context?
>
> **A4**: We thank the reviewer for this insightful question. A unified evaluation metric provides three critical benefits in the context of complex trip planning.
>
> 1.  **Bridging the gap between synthetic benchmarks and real-world utility**. Existing research often over-optimizes for "Constraint Pass Rate" on synthetic templates, a metric that correlates poorly with actual user satisfaction. By unifying strict constraints with soft quality alignment, our reward, which is validated against human experts, serves as a trusted proxy for real-world performance. This shifts the community focus from building mere logical solvers to developing robust assistants capable of handling the ambiguity of authentic, free-form requests.
>
> 2.  **Facilitating fine-grained, holistic comparisons**. Complex planning involves trade-offs (e.g., balancing budget vs. comfort) that binary success/failure metrics cannot capture. TripScore provides a continuous, standardized yardstick that quantifies these "soft" aspects of planning quality. This allows for nuanced, head-to-head comparisons and resolves the ambiguity of weighing disparate sub-metrics.
>
> 3.  **Serving as a granular optimization objective for RL**. Perhaps most importantly, unifying feasibility and quality metrics into a single scalar reward creates a clear optimization target. This enables the community to move beyond prompt engineering and test-time search, facilitating the application of Reinforcement Learning (e.g., PPO, GRPO) to directly align models with complex planning objectives, as demonstrated by the significant performance gains in our experiments.
>
>
> **Q3**: Compared with ChinaTravel, which focuses on NeSy reasoning, what type of approach do you personally find more promising for advancing travel planning tasks, and what are your key insights?
>
> **A5**: Based on our comparative analysis, we believe **End-to-End Reinforcement Learning** (e.g., GRPO) is more promising than rigid Neuro-Symbolic (NeSy) pipelines or pure test-time search for real-world travel planning. Our conclusion is supported by four key insights regarding the trade-offs between flexibility, efficiency, and model capacity:
>
> 1.  **The Rigidity of Neuro-Symbolic Solvers (The "Constraint Paradox")**. While NeSy approaches (like ChinaTravel or TTG) achieve near-perfect Constraint Pass Rates (99%), they suffer from exceptionally low Delivery Rates and Rewards. This highlights a "Constraint Paradox": symbolic solvers treat user preferences as rigid logical rules, leading to "over-constrained" states where no solution is found if perfect conditions aren't met. In contrast, end-to-end models handle "soft" constraint satisfaction, finding viable trade-offs that yield executable plans even under ambiguity—a critical requirement for real-world utility.
>
> 2.  **Diminishing Returns of Test-Time Compute**. While test-time methods (e.g., LLM-Modulo, HyperTree) improve feasibility, they incur prohibitive latency costs. As shown in Table 2, while LLM-Modulo with GPT-4o achieves the highest reward, it triples the inference time compared to Direct prompting (62.39s vs. 21.36s). The marginal gain in reward often fails to justify the linear or exponential increase in latency, scaling poorly for real-time user applications.
>
> 3.  **Ineffectiveness of Complex Scaffolding for Small Models**. We observe a striking disparity in how model size interacts with test-time strategies. On smaller models like Qwen3-8B, advanced methods like HyperTree actually degrade performance (Reward drops from -1.00 to -2.62) compared to direct prompting. It appears that smaller models lack the robust instruction-following capabilities needed to navigate complex reasoning scaffolding. Effective planning via search seems to be an emergent ability requiring a stronger base reasoner; for smaller models, RL provides a much more effective guidance mechanism.
>
> 4.  **Internalizing Reasoning via RL Fine-tuning**. We find that RL fine-tuning offers the optimal path forward by "internalizing" complex planning reasoning into parametric knowledge, achieving a superior balance of quality and efficiency.
>
> - Feasibility. On Qwen3-8B, our GRPO fine-tuning achieves a Real-world Reward of 2.04, significantly outperforming Direct prompting (-1.00) and even surpassing the proprietary direct prompting GPT-4o baseline (1.61), all while maintaining ultra-low latency (<10s).
>
> - Quality. Beyond reward scores, human expert evaluation confirms genuine quality gains. On Qwen3-14B, GRPO reduces the expert preference for GPT-4o from a dominant 78.5% (vs. Direct) to a statistical tie of 51.3% (vs. GRPO).
>
> This demonstrates that parameter adaptation is far more compute-efficient than external solvers or test-time search. It effectively bridges the capability gap, enabling small open-source models to match leading proprietary models in both feasibility and human-perceived quality.

---

> ### Author Response · Authors · 2025-11-25
> **(4/4)**
>
> **Q4**: Why was the ReAct baseline not included in the experiments?
>
> **A6**: We thank the reviewer for this suggestion and have added the **ReAct** baseline to Table 2.
>
> Key Findings: While ReAct generally improves performance over Direct prompting, it remains suboptimal compared to our proposed RL approach in terms of both efficacy and efficiency, particularly for smaller models.
>
> 1. **Efficacy Gap (RL > ReAct)**. On the Real-world track (Qwen3-14B), ReAct achieves a Reward of 0.66, which is a notable improvement over Direct (-0.24). However, it still significantly lags behind our GRPO fine-tuning (Reward 2.20). This confirms that iteratively acting via external tools is less effective than "internalizing" the planning policy via RL.
>
> 2. **Efficiency Penalty (High Latency)**. ReAct incurs a substantial computational cost. For Qwen3-14B, ReAct takes 31.75s per query, which is 2.3x slower than the GRPO model (14.04s).
>
> The inclusion of ReAct further validates our core hypothesis. While test-time scaffolding (like ReAct or Modulo) helps, RL fine-tuning remains the Pareto-optimal solution, delivering better plan quality with much lower latency.
>
> | Method |  Synthetic |  |  |  |   | Real-world |  |  |  |    |
> |-------- |-----------|--|--|--|------------|--|--|--|--|--|
> |  | DR↑ | CPR↑ | CondR↑ | Reward↑ | Time↓ | DR↑ | CPR↑ | CondR↑ | Reward↑ | Time↓ |
> | Direct |  |  |  |  |  |  |  |  |  |  |  |
> | GPT-4o | 77.23 | 93.73 | 2.87 | 1.40 | 20.71 | 76.71 | 80.61 | 3.73 | 1.61 | 21.36 |
> | GPT-4.1-Mini | 75.69 | 91.43 | 2.89 | 1.27 | 13.96 | 75.34 | 68.49 | 3.74 | 1.19 | 14.22 |
> | DSV3 | 73.92 | 96.00 | 2.87 | 1.26 | 27.61 | 75.78 | 78.10 | 3.66 | 1.44 | 25.03 |
> | Qwen3-8B | 27.14 | 94.98 | 3.39 | -1.31 | 9.37 | 39.73 | 56.30 | 3.61 | -1.00 | 10.48 |
> | Qwen3-14B | 39.93 | 88.70 | 3.05 | -0.72 | 12.35 | 51.59 | 67.26 | 3.49 | -0.24 | 13.15 |
> | Qwen3-32B | 42.32 | 95.13 | 2.98 | -0.53 | 19.33 | 60.27 | 71.86 | 3.88 | 0.49 | 19.83 |
> | Test-Time Compute |  |  |  |  |  |  |  |  |  |  |
> | CoT GPT-4o | 81.65 | 93.96 | 2.88 | 1.66 | 20.48 | 82.19 | 80.55 | 3.70 | 1.92 | 20.14 |
> | CoT Qwen3-8B | 22.53 | 93.47 | 3.39 | -1.61 | 13.77 | 36.07 | 51.89 | 3.62 | -1.24 | 13.51 |
> | CoT  Qwen3-14B | 30.85 | 83.43 | 2.89 | -1.33 | 17.59 | 45.79 | 70.40 | 3.68 | -0.44 | 17.27 |
> | **ReAct GPT-4o** | 82.80 | 90.42 | 2.89 | 1.65 | 32.14 | 78.54 | 74.99 | 3.74 | 1.56 | 39.60 |
> | **ReAct Qwen3-8B** | 31.10 | 93.14 | 2.89 | -1.23 | 18.98 | 40.10 | 72.68 | 3.66 | -0.73 | 21.88 |
> | **ReAct Qwen3-14B** | 54.89 | 81.79 | 2.88 | -0.06 | 26.16 | 60.73 | 80.45 | 3.76 | 0.66 | 31.75 |
> | LLM-Modulo GPT-4o | 84.38 | 92.91 | 2.91 | 1.81 | 44.78 | 90.86 | 89.95 | 3.63 | 2.69 | 62.39 |
> | LLM-Modulo Qwen3-8B | 44.33 | 90.32 | 2.94 | -0.49 | 34.17 | 50.22 | 72.72 | 3.65 | -0.16 | 40.23 |
> | LLM-Modulo Qwen3-14B | 49.54 | 85.42 | 2.89 | -0.29 | 64.38 | 56.62 | 81.56 | 3.40 | 0.27 | 54.14 |
> | HyperTree GPT-4o | 81.51 | 89.39 | 2.89 | 1.55 | 49.10 | 84.01 | 70.11 | 3.73 | 1.72 | 45.96 |
> | HyperTree Qwen3-8B | 14.67 | 85.75 | 2.94 | -2.19 | 31.31 | 4.56 | 100.00 | 3.80 | -2.62 | 35.92 |
> | HyperTree Qwen3-14B | 27.25 | 66.92 | 2.86 | -1.66 | 34.21 | 42.01 | 63.04 | 3.85 | -0.72 | 41.71 |
> | Neural-Symbolic |  |  |  |  |  |  |  |  |  |  |
> | TTG GPT-4o | 37.10 | 99.38 | 3.60 | -0.56 | 2.21 | 12.78 | 100.00 | 3.49 | -2.17 | 2.03 |
> | NESY GPT-4o | 57.96 | 98.74 | 3.25 | 0.60 | 77.92 | 1.82 | 100.00 | 3.04 | -2.89 | 46.44 |
> | Qwen3-8B | 47.37 | 98.04 | 2.86 | -0.25 | 62.58 | 2.73 | 100.00 | 3.22 | -2.83 | 42.12 |
> | Fine-Tuning |  |  |  |  |  |  |  |  |  |  |
> | SFT Qwen3-8B | 74.02 | 91.23 | 2.88 | 1.17 | 9.00 | 77.16 | 82.84 | 3.46 | 1.53 | 11.85 |
> | SFT Qwen3-14B | 82.80 | 93.41 | 2.86 | 1.70 | 12.69 | 85.38 | 76.47 | 3.56 | 1.89 | 15.34 |
> | RFT Qwen3-8B | 74.49 | 93.66 | 2.90 | 1.26 | 9.77 | 82.19 | 81.11 | 3.53 | 1.82 | 12.21 |
> | RFT Qwen3-14B | 84.75 | 93.44 | 2.88 | 1.83 | 15.13 | 84.02 | 77.71 | 3.52 | 1.83 | 14.43 |
> | GRPO Qwen3-8B | 75.59 | 90.05 | 2.91 | 1.25 | 8.97 | 88.31 | 76.72 | 3.53 | 2.04 | 9.67 |
> | GRPO Qwen3-14B | 84.91 | 94.56 | 2.89 | 1.87 | 15.34 | 90.27 | 78.50 | 3.52 | 2.20 | 14.04 |

---

> > ### Comment · Reviewer_h4xw · 2025-11-28
> >
> > I think the author has well resolved my confusion, and I reserve my positive view of this paper.

---

### Official Review · Reviewer_bfBK · 2025-11-01

**Soundness:** 2
**Presentation:** 2
**Contribution:** 2
**Rating:** 4
**Confidence:** 4

**Summary:**

This paper considers the travel planning, proposes TripScore, and a unified reward for evaluating multi-constraint travel itineraries. This reward function integrates the rule-based evaluation and LLM-as-judge. Experiments compare direct prompting, test-time reasoning, neuro-symbolic approaches, and fine-tuning including GRPO, reporting gains in delivery rate, commonsense pass rate, and the unified reward, with analyses of error types and trip-duration sensitivity.

**Strengths:**

1. The problem is practically relevant and the authors implement a comprehensive evaluation workflow that attempts to separate hard feasibility from softer quality criteria.
2. The paper provides careful engineering details, ablations over trip duration, error breakdowns, and an expert-annotation study to partially validate the reward.

**Weaknesses:**

1. The reliability of the unified reward is limited for its intended purpose. Agreement with human experts is only 60.75%. Because the reward is heavily shaped by the gating and by narrow penalty ranges, the final score collapses much of the variation into a few bands that correlate strongly with format and commonsense feasibility, undermining its claim to measure fine-grained quality beyond validity. From my perspective, a unified reward design is a methodological requirement rather than a benchmark requirement. However, the RL algorithm based on this reward proposed by the authors does not seem to have achieved a significant improvement compared to training-free algorithms.
2. In summary, from a product development perspective, if the proposed reward design ultimately yields an effective RL solution, then I would consider this work to be more beneficial than TravelPlanner. However, from Benchamrk's current perspective, the incremental benefits of this work compared to TravelPlanner are limited.
3. The use of charts is too similar to that of TravelPlanner, especially Figure 1, which is almost identical. Table 2 and Figure 4 are also very similar in design. This level of similarity necessitates special attribution to the figure as originating from TravelPlanner.

**Questions:**

1. Compared to TravelPlanner, what are the core contributions of this article?
2. What impact does the author believe the presented unified reward will have on the travel planning community?
1. How do you address the training–evaluation mismatch that risks Goodhart’s law, given RL/RFT use only the rules-based reward while the test metric includes LLM-scored components?

---

> ### Author Response · Authors · 2025-11-25
> **(1/3)**
>
> We sincerely thank you for your meticulous review and insightful suggestions, which have been instrumental in refining our manuscript.
>
> **W1-1&Q3**: The reliability of the unified reward and the potential risk of Goodhart's law due to the mismatch between training rewards and evaluation metrics.
>
> **A1**: We thank the reviewer for this insightful critique. To address these questions, we have conducted a rigorous **additional human expert evaluation** in Table 3. We compare the generated plans head-to-head to assess the quality produced by different method. We paired the outputs of our trained models (Qwen3-14B) against the leading proprietary model (GPT-4o) and asked human experts to blindly select the superior plan.
>
> As shown in the table, optimizing the unified reward via GRPO does not merely improve the feasibility; it also improve the plan quality. Under "Expert Selection - Real-world," the baseline Qwen3-14B (Direct) loses heavily to GPT-4o (GPT-4o Win Rate: 78.5%). After RL training (GRPO), the model reaches near-parity with GPT-4o (GPT-4o Win Rate: 51.3%).
>
> If Goodhart’s law were dominating, we would see high TripScore reward but low expert preference. Instead, we observe that the TripScore reward acts as a robust proxy for paln quality, thereby effectively guiding the model toward human-aligned performance.
>
>
> | Test Set Pair (A vs B) | A Win Rate | 95% CI | p-value |
> |------------------------|------------|--------|---------|
> | **TripScore Reward – Real-world** | | | |
> | Direct (GPT-4o) vs Direct (Qwen3-14B) | 0.744 | [0.589, 0.854] | 0.004 |
> | Direct (GPT-4o) vs GRPO (Qwen3-14B) | 0.522 | [0.486, 0.557] | 0.245 |
> | **Expert Selection – Real-world** | | | |
> | Direct (GPT-4o) vs Direct (Qwen3-14B) | 0.785 | [0.671, 0.829] | 0.002 |
> | Direct (GPT-4o) vs GRPO (Qwen3-14B) | 0.513 | [0.478, 0.548] | 0.492 |
>
> **W1-2**: the reward is heavily shaped by the gating and by narrow penalty ranges. the final score collapses much of the variation into a few bands that correlate strongly with format and commonsense feasibility, undermining its claim to measure fine-grained quality beyond validity.
>
> **A2**: We appreciate this insightful critique. To prove that TripScore measures fine-grained quality beyond mere validity, we conducted a Controlled Pairwise Comparison in Table 3 specifically targeting the reviewer's concern.
>
> 1. **Controlled Experiment: Isolating Quality from Feasibility**. To disentangle quality from feasibility, we filtered the test set to include only queries where both competing plans passed the feasibility check. This neutralizes the "gating" effect mentioned by the reviewer, leaving only the "quality" component as the differentiator. We then compared the plans generated by the Base Model (Qwen3-14B Direct) vs. the RL-tuned Model (Qwen3-14B GRPO) using both TripScore Reward and Blind Expert Voting.
>
> 2. **Results: High Discriminative Power for Quality**. When feasibility is held constant, TripScore reveals **significant quality variance and aligns robustly with human experts** (see Table 3). Our results demonstrate a remarkable alignment between TripScore and human expert judgment, particularly on the challenging Real-world track.
>
> - For the Direct (Qwen3-14B) vs Direct (GPT-4o) model, TripScore predicts a 74.4% win rate for GPT-4o, closely matching the expert's 78.5%.
>
> - For the SFT(Qwen3-14B) vs Direct (GPT-4o) model, TripScore predicts a 65.9% win rate for GPT-4o, closely matching the expert's 61.4%.
>
> - For the RFT(Qwen3-14B) vs Direct (GPT-4o) model, TripScore predicts a 55.2% win rate for GPT-4o, closely matching the expert's 56.1%.
>
> - As we optimize the model via GRPO, TripScore indicates a narrowing gap (GPT-4o win rate drops to 52.2%). Crucially, experts confirm this exact trend, with the GPT-4o win rate dropping to 51.3%.
>
> This synchronized convergence, where the metric and experts agree not just on the ranking but on the magnitude of the gap, validates TripScore as a highly reliable proxy for human utility.
>
> | Test Set Pair (A vs B) | A Win Rate | 95% CI | p-value |
> |------------------------|------------|--------|---------|
> | **TripScore Reward – Real-world** | | | |
> | Direct (GPT-4o) vs Direct (Qwen3-14B) | 0.744 | [0.589, 0.854] | 0.004 |
> | Direct (GPT-4o) vs SFT (Qwen3-14B) | 0.659 | [0.505, 0.784] | 0.032 |
> | Direct (GPT-4o) vs RFT (Qwen3-14B) | 0.552 | [0.375, 0.716] | 0.711 |
> | Direct (GPT-4o) vs GRPO (Qwen3-14B) | 0.522 | [0.486, 0.557] | 0.245 |
> | **Expert Selection – Real-world** | | | |
> | Direct (GPT-4o) vs Direct (Qwen3-14B) | 0.785 | [0.671, 0.829] | 0.002 |
> | Direct (GPT-4o) vs SFT (Qwen3-14B) | 0.614 | [0.572, 0.654] | 0.008 |
> | Direct (GPT-4o) vs RFT (Qwen3-14B) | 0.561 | [0.409, 0.682] | 0.636 |
> | Direct (GPT-4o) vs GRPO (Qwen3-14B) | 0.513 | [0.478, 0.548] | 0.492 |

---

> ### Author Response · Authors · 2025-11-25
> **(2/3)**
>
> **W1-3**: RL Effectiveness
>
> **A3**: We respectfully disagree with the assessment that RL shows "limited improvement." Our results confirm that RL is superior to training-free methods across three critical dimensions:
>
> 1. **The Efficiency & Scalability Gap (Product Perspective)**. While training-free methods (e.g., LLM-Modulo) can improve the reward score, they are impractical for deployment. RL is the only viable path to enable high-quality planning on efficient, deployable models (14B), matching GPT-4o's utility at a fraction of the cost and latency.
>
> - Latency. Test-time search incurs prohibitive latency (62.39s for LLM-Modulo GPT-4o) compared to RL (9.67s for GRPO Qwen-8B). It is a 6x speedup.
>
> - Capacity Threshold. Test-time methods fail on smaller models. On Qwen-8B, adding search (HyperTree) actually degrades reward (-1.00 to -2.62), whereas RL drastically improves it.
>
>
> 2. **The "Limited Improvement" of RL**.
>
> - The improvement in feasibility is statistically significant, not marginal. On the synthetic dataset (Table 2), RL (Qwen3-14B) boosts the Delivery Rate from 39.93% to 84.91%  and Commonsense Pass Rate from 88.70% to 94.56% compared to the Direct baseline. This transforms a barely usable model into a robust planner.
>
> - The improvement in plan quality is also notable. As established in **A1**, RL is the key driver for quality alignment. From Table 3, RL (GRPO) reduces the "Expert Win Rate" gap with GPT-4o by 27.2 percentage points (from 78.5% loss to 51.3% parity), significantly outperforming Supervised Fine-Tuning (SFT), which still loses to GPT-4o (61.4% win rate).
> This confirms that our RL algorithm effectively bridges the chasm between open-source models and proprietary leading models, a feat that training-free methods failed to achieve on this model scale.
>
>
> **W2**: Methodological Contribution
>
> **A4**: We respectfully clarify that TripScore is a cohesive benchmark where the Reward Design and Dataset are closely linked, not separate methodological add-ons.
>
> 1. **Data-Driven Evaluation Protocol**. The unified reward is not an arbitrary methodological choice. It is a direct consequence of our behavior-grounded dataset. As our logs reveal, 39.3% of users select only destinations and duration, and 38.3% input vague, free-text requests. For such data, a traditional "Pass/Fail" metric with multiple logical constraints is insufficient. Therefore, without excessive logical constraints, it is necessary to evaluate feasibility along with plan quality on this real-world data.
>
> 2. A benchmark contributes two things: What to solve (Data) and How to measure success (Evaluation).
>
> - Data Contribution: We introduce a dual-track dataset mirrored from our real-world log analysis. The synthetic track (TripScore-S) targets the 39.3% of users providing minimal constraints (destinations and duration), while the real-world track (TripScore-R) captures the 38.3% submitting free-form user requests.
>
> - Evaluation Contribution: We propose a two-stage evaluation framework: a minimal feasibility gate, followed by a weighted reward for quality assessment.
>
> Separating them would undermine the benchmark's utility. TripScore packages both to ensure the community has a complete, reproducible standard for high quality planning.
>
>
> **W3**: The use of charts is too similar to that of TravelPlanner, especially Figure 1, which is almost identical. Table 2 and Figure 4 are also very similar in design. This level of similarity necessitates special attribution to the figure as originating from TravelPlanner.
>
> **A5**: We appreciate the reviewer's attention to visual originality. We take this concern seriously and have extensively revised our visual elements to ensure they distinctly reflect TripScore's unique contributions.
>
> 1. Complete Redesign of Figure 1. We have completely redesigned Figure 1 from scratch. The new figure adopts an arrow-based workflow diagram that specifically highlights our novel "Unified Reward" pipeline and dataset construnction process. This new structure visually and conceptually distances our work from prior layouts.
>
> 2. Differentiation in Table 2 & Figure 4.
>
> - Table 2: To explicitly distinguish our contribution, we have highlighted all novel constraints in blue. This visual cue allows readers to instantly identify the unique evaluation dimensions introduced by TripScore that are not present in existing benchmarks.
>
> - Figure 4: We have also redrawn Figure 4 with a new color scheme to prevent any potential confusion, ensuring that the presentation of our experimental results is entirely original.
>
> We believe these changes fully address the concern regarding visual similarity.

---

> ### Author Response · Authors · 2025-11-25
> **(3/3)**
>
> **Q1**: Compared to TravelPlanner, what are the core contributions of this article?
>
> **A6**: We thank the reviewer for giving us the opportunity to clarify our positioning.
>
> 1. **Exposing the Failure of Symbolic Solvers**. We identify a critical flaw in existing constraint-based benchmarks like TravelPlanner, that they unintentionally favor neuro-symbolic approaches (e.g., TTG, NESY) that excel at rigid logical puzzles but fail in realistic scenarios. Our experiments demonstrate that while these solvers achieve high pass rate on logical constraints, they struggle significantly with real-world queries (low Reward). TripScore quantitatively demonstrates that once queries move beyond hard constraints, neuro-symbolic methods struggle to generalize to real-world scenarios.
>
> 2. **Shifting from Artificial Constraints to User Intent**. TravelPlanner relies on synthetic queries with rigid constraints. In contrast, our analysis of real-world user logs reveals a stark difference: users rarely choose explicit, predefined constraints. We found that 39.3% of users select only destinations & duration, and 38.3% input vague, free-text requests. Consequently, a constraint-based evaluator is insufficient. TripScore introduces a two-stage evaluation (minimal Feasibility Check + Unified Quality Score) to capture the nuance of plan quality.
>
> 3. **Unifying Reward for RL**. Unlike TravelPlanner’s discrete evaluation, TripScore proposes a Unified Reward mechanism that fuses feasibility, quality, and preference into a granular scalar signal. This is a key enabler for End-to-End Reinforcement Learning. As evidenced by our GRPO results, this granular score allows models to be optimized both for plan feasibility and quality, bridging the gap towards leading proprietary performance.
>
> 4. **Expert Validation**. Most critically, as shown in the Table below, TripScore is backed by rigorous human expert validation. While recent works (TripTailor, RealTravel) rely solely on LLM-as-a-judge, our unified score is calibrated against and aligned with human expert judgment, ensuring that our metric reflects the real-world utility rather than just model-based hallucinations.
>
> | Benchmark | Query Type | Query Source | Quality Evaluation | Output Form | Expert Validation |
> |-----------|------------|--------------|--------------------|-------------|-------------------|
> | TravelPlanner (Xie et al., 2024) | CB Template | Synthetic | ✗ | Pass rate | ✗ |
> | Trip Planning (Zheng et al., 2024) | CB Template | Synthetic | ✗ | Pass rate | ✗ |
> | ChinaTravel (Shao et al., 2024a) | CB Free-Text | AI-Generated  | Rule | Pass rate + Per-Dim ranking | ✗ |
> | TripTailor (Wang et al., 2025) | CB Free-Text | AI-Generated  | LLM | Pass rate + Surpass rate | ✗ |
> | RealTravel (Shao et al., 2025) | CB Free-Text | AI-Generated  |  LLM | Pass rate + Preference rate | ✗ |
> | **TripScore-S (ours)** | **CB Template** | **Real logs (behav.)** | **Rule + LLM** | **Unified score** | **✓** |
> | **TripScore-R (ours)** | **Free-form req.** | **Real logs (req.)** | **Rule + LLM** | **Unified score** | **✓** |
>
>
> **Q2**: What impact does the author believe the presented unified reward will have on the travel planning community?
>
> **A7**: We believe that the unified score will serve as a shift from "Constraint Satisfaction" to "Quliaty Optimization" in the travel planning community.
>
> 1. **Bridge the gap between synthetic benchmark and real-world utility**. Current research often over-optimizes for "Constraint Pass Rate" on synthetic templates, which correlates poorly with real-world utility. Our reward, validated against human experts and designed for authentic user intents (e.g., free-form requests), provides a trusted proxy for real-world performance. This encourages the community to develop planners that are not just logical solvers, but robust assistants capable of handling ambiguity.
>
> 2. **Standardizing "Soft" Evaluation**. We offer a flexible, verified metric that goes beyond binary success/failure. Previously, evaluating the "soft" aspects of a plan (e.g., pacing, diversity) was subjective and inconsistent. TripScore offers a standardized, reproducible yardstick for these nuances, which is validated by human expert. This facilitates rigorous, fine-grained comparisons between methods, accelerating progress in the subtler, human-centric dimensions of AI planning.
>
> 3. **Enabling the next generation of Learning-based Planners (RL)**. Our reward can enable end-to-end RL. By unifying disparate feasibility and quality metrics into a single, granunal reward (TripScore), we provide the community with a standard optimization target. This allows future research to move beyond prompt engineering and test-time search, enabling the application of Reinforcement Learning to directly align models with complex planning objectives, as demonstrated by our GRPO results.

---

### Note · Authors · 2025-12-01

I have read and agree with the venue's withdrawal policy on behalf of myself and my co-authors.